# Coresets for Near-Convex Functions

**Murad Tukan**
muradtuk@gmail.com

**Alaa Maalouf**
alaamalouf12@gmail.com

**Dan Feldman**
dannyf.post@gmail.com

The Robotics and Big Data Lab,
Department of Computer Science,
University of Haifa,
Haifa, Israel

## Abstract

Coreset is usually a small weighted subset of $n$ input points in $\mathbb{R}^d$, that provably approximates their loss function for a given set of queries (models, classifiers, etc.). Coresets become increasingly common in machine learning since existing heuristics or inefficient algorithms may be improved by running them possibly many times on the small coreset that can be maintained for streaming distributed data. Coresets can be obtained by sensitivity (importance) sampling, where its size is proportional to the total sum of sensitivities. Unfortunately, computing the sensitivity of each point is problem dependent and may be harder to compute than the original optimization problem at hand. We suggest a generic framework for computing sensitivities (and thus coresets) for wide family of loss functions which we call near-convex functions. This is by suggesting the $f$-SVD factorization that generalizes the SVD factorization of matrices to functions. Example applications include coresets that are either new or significantly improves previous results, such as SVM, Logistic regression, M-estimators, and $\ell_z$-regression. Experimental results and open source are also provided.

## 1   Introduction

In common machine learning problems, we are given a set of input points $P \subseteq \mathbb{R}^d$ (training data), and a loss function $f : P \times \mathbb{R}^d \to [0, \infty)$, where the goal is to solve the optimization problem of finding a query (model, classifiers, centers) $x^*$ that minimizes the sum of fitting errors $\sum_{p \in P} f(p, x)$ over every query $x$ in a given (usually infinite) set. For example, in $k$-median (or $k$-mean) clustering, each query is a set of $k$ centers and the loss function is the distance (or squared distance) of a point to its nearest center. In linear regression or SVM, every input point includes a label, and the loss function is the fitting error between the classification of $p$ via a given query to the actual label of $p$. Empirical risk minimization (ERM) may be used to generalize the result from train to test data.

**Modern machine learning.** In practice, many of these optimization or learning problems are usually hard even to approximate. Instead, practical heuristics with no provable guarantees may be used to solve them. Even for well understood problems, which have close optimal solution, such as linear regression or classes of convex optimization, in the era of big data we may wish to maintain the solution in other computation models such as: streaming input data ("on-the-fly") that provably uses small memory, parallel computations on distributed data (on the cloud, network or GPUs) as well as deletion of points, constrained optimization (e.g. sparse classifiers). Cross validation [34] or hyper-parameter tuning techniques such as AutoML [30, 32] need to evaluate many queries for different subsets of the data, and different constraints.

**Coresets.** One approach is to redesign existing machine learning algorithms for faster, approximate solutions and these new computation models. A different approach that is to use data summarization techniques. *Coresets* in particular were first used to solve problems in computational geometry [1] and

got increasing attention [3, 4, 5, 6, 8, 20, 26, 27, 38] over the recent years; see surveys in [23, 47, 50]. Informally, coreset is a small weighted subset of the input points (unlike e.g. sketches, or dimension-reduction techniques) that approximates the loss of the input set $P$ for *every* feasible query $x$, up to a provable bound of $1 \pm \varepsilon$ for a given error parameter $\varepsilon \in (0, 1)$. The size of the coreset is usually polynomial in $1/\varepsilon$ but independent or near-logarithmic in the size of the input. Since such a coreset approximates every query (and not just the optimal one), it supports constraint optimization, and the above computation models using merge-and-reduce trees; see details in [23]. Moreover, coresets may be computed in time that is near-linear in the input, even for NP-hard optimization problems. Existing heuristic or inefficient algorithms may then be applied many times on the small coreset to obtain improved or faster models in such cases.

Example coresets in machine learning include *SVM* [33, 57, 58, 59, 60], $\ell_z$-regression [18, 21, 54], clustering [2, 16, 24, 31, 37, 42, 53], logistic regression [35, 47], LMS solvers and *SVD* [28, 44, 45, 52], where all of these works have been dedicated to suggest a coreset for a specific problem.

**A generic framework** for constructing coresets was suggested in [25, 40]. It states that, with high probability, non-uniform sampling from the input set yields a coreset. Each point should be sampled i.i.d. with a probability that is proportional to its importance or sensitivity, and assigned a multiplicative weight which is inverse proportional to this probability, so that the expected original sum of losses over all the points will be preserved. Here, the sensitivity of an input point $p \in P$ is defined to be the maximum of its relative fitting loss $s(p) = f(p, x)/\sum_{q \in P} f(q, x)$ over every possible query $x$. The size of the coreset is near-linear in the total (sum) $t$ of these sensitivities; see Theorem 3 for details. It turns out in the recent years that many classical and hard machine learning problems [7, 43, 55] have total sensitivity that is near-logarithmic or independent of the input size $|P|$ which implies small coresets via sensitivity sampling.

**Paper per problem.** The main disadvantage of this framework is that the sensitivity $s(p)$, as defined above, is problem dependent: namely on the loss function $f$ and the feasible set of queries. Moreover, maximizing $s(p) = f(p, x)/\sum_{q \in P} f(q, x)$ is equivalent to minimizing the inverse $\sum_{q \in P} f(q, x)/f(p, x)$. Unfortunately, minimizing the enumerator is usually the original optimization problem which motivated the coreset in the first place. The denominator may make the problem harder, in addition to the fact that now we need to solve this optimization problem for each and every input point in $P$. While approximations of the sensitivities usually suffice, sophisticated and different approximation techniques are frequently tailored in papers of recent machine learning conferences for each and every problem.

## 1.1 Problem Statement

To this end, the goal of this paper is to suggest a framework for sensitivity bounding of a *family* of functions, and not for a specific optimization problem. This approach is inspired by convex optimization: while we do not have a single algorithm to solve any convex optimization, we do have generic solutions for family of convex functions. E.g., linear programming, Semi-definite programming, and so on.

We choose the following family of near-convex loss functions, with example supervised and unsupervised applications that include support vector machines, logistic regression, $\ell_z$-regression for any $z \in (0, \infty)$, and functions that are robust to outliers. In the Supplementary Material we suggest a more generalized version that handles a bigger family of functions; see Definition 13, and hope that this paper will inspire the research of more and larger families.

**Definition 1** (Near-convex functions)**.** *Let $P \subseteq \mathbb{R}^d$ be a set of $n$ points, and let $f : P \times \mathbb{R}^d \to [0, \infty)$ be a loss function. We call $f$ a near-convex function if there are a convex loss function $g : P \times \mathbb{R}^d \to [0, \infty)$ (see Definition 12 at Supplementary Material), a function $h : P \times \mathbb{R}^d \to [0, \infty)$, and a scalar $z > 0$ satisfying:*

*(i) There exist $c_1, c_2 > 0$ such that for every $p \in P$, and $x \in \mathbb{R}^d$,*

$$c_1 \left( g(p, x)^z + h(p, x)^z \right) \le f(p, x) \le c_2 \left( g(p, x)^z + h(p, x)^z \right).$$

*(ii) For every $p \in P$, $x \in \mathbb{R}^d$ and $b > 0$, we have $g(p, bx) = b \cdot g(p, x)$.*

*(iii) For every $p \in P$ and $x \in \mathbb{R}^d$, we have $\frac{h(p,x)^z}{\sum_{q \in P} h(q,x)^z} \le \frac{2}{n}$.*

*(iv) The set $\mathcal{X}_g = \left\{ x \in \mathbb{R}^d \middle| \sum_{p \in P} g(p,x)^{\max\{1,z\}} \leq 1 \right\}$ is centrally symmetric, i.e., for every $x \in \mathcal{X}_g$ we have $-x \in \mathcal{X}_g$, and there exist $R, r \in (0, \infty)$ such that $B(0_d, r) \subset \mathcal{X}_g \subset B(0_d, R)$, where $B(0_d, y)$ denotes a ball of radius $y > 0$, centered at $0_d$.*

*We denote by $\mathcal{F}$, the union of all functions $f$ with the above properties.*

**The intuition behind Definition 1.**  Properties (i)-(iii) are used to reduce the problem to dealing with a "simpler" pair of functions where the first is a convex function "g" that is linear in its argument $x$ and the second function "h" being independent of the input points. Property (iv) ensures that the ellipsoid which encloses the level set of $g$ (the convex function) exists and is centered at the origin to avoid dealing with the center. By combining the properties associated with the level set of $g$ (the convex function) and Properties (i)-(iv), we manage to bound the loss function from above and below by the mahalanobis distance with respect to the enclosing ellipsoid. This is due to the fact that the level set encloses a contracted version of the ellipsoid which encloses the level set of $g$.

We are interested in a generic algorithm that would get a set of input points, and a loss function as above, and compute a sensitivity for each point, based on the parameters of the given loss function. In addition, we wish to use worst-case analysis and prove that for every input the total sensitivity (and thus size of coreset) would be small, depending on the "hardness" of the loss function that is encapsulated in the above parameters $z, R$, etc.

## 2   Related Work

**Logistic Regression.** A coreset construction algorithms for the problem of logistic regression were suggested by [35], [56], and [47]. All of these works handled variations of the problem, e.g., they all lack the incorporation of the bias term (intercept) in their loss function. Specifically speaking, both [35] and [47] didn't account for the regularization term and its parameter. Furthermore, the coreset's size established by [47], was dependant on the structure of the input data. As for [56], the coreset only succeed for a small subset of queries (a ball in $\mathbb{R}^d$ of radius $r$, where the coreset's size is near linear in $r$). Contrary to previous works, our coreset approximates the logistic regression loss function including the bias parameter (intercept) and the regularization term for every possible query. This is the loss function that is usually used in practice, e.g., see Sklearn library in [49]. Finally, our coreset's size is independent of the structure of the data.

**SVM.** [11, 57, 58] addressed the problem of coreset construction for SVM, yet they used squared hinge loss to enforce the SVM cost function to be strongly convex. At [60], the coreset is constructed with respect to the hinge loss which most used form of SVM in practice (see Sklearn library at [49]). However for the coreset to be constructed, a (sub-)optimal solution was required for the problem itself. In addition, the coreset size depended heavily on on the ratio between the variance of each class of points. In this paper, we also address a coreset with respect to the hinge loss, yet we don't require any (sub-)optimal solution to construct the coreset, and our coreset's size depends on the ratio between the number of points of each class (see Corollary 9).

$\ell_z$**-Regression.** A notable line of work [10, 18, 21, 54, 65] addressed the construction of coresets and sketches in this area, however, all such papers addressed the case of $z \geq 1$. Most of these works used tools similar to the well-conditioned basis which was first suggested at [21] to compute such coresets. Intuitively it can be thought of as a generalization of the *SVD* factorization of an input set with respect to the loss function of $\ell_z$-regression for any $z \geq 1$. In our framework we generalize this factorization in order to compute coresets for the near-convex functions. To our knowledge, we suggest the first coreset for the problem of $\ell_z$-regression for any $z \in (0, 1)$.

**Outlier resistant functions (similar to $M$-estimators).** Among such functions, is the $\ell_z$-regression for any $z \in (0, 1]$ that is mentioned above, *Huber* loss function [13], *Tukey* loss functions [12], and many more [14]. However, to our knowledge, we present the first coreset for the problem formulation which is given at Corollary 10.

## 3   Our contribution

In this paper, we suggest an $\varepsilon$-coreset construction algorithm with respect to any near-convex function. Specifically speaking, we provide:

(i) A generalization of the well conditioned bases of [21] to a broader family of functions, i.e., not just for $\ell_z$-Regression problems where $z \geq 1$. This informally describes a factorization of the input data with respect to a given near-convex loss function. We call such factorization the $f$-SVD of $P$ (see Definition 4).

(ii) A framework for bounding the sensitivity of each point in an input set with respect to any near-convex function. The heart of the framework relies on computing the $f$-SVD factorization described in (i); see Lemma 5 and Algorithm 1.

(iii) By (ii), we provide the first $\varepsilon$-coreset for the problem of $\ell_z$-regression where $z \in (0, 1)$, and the first $\varepsilon$-coreset for certain outlier resistant functions. We also unify existing works of coreset construction for the problems of logistic regression and SVM; see Section 6.

(iv) Experimental results on real-world and synthetic datasets for common machine learning solvers (supported by our framework) of Scikit-learn library [49], assessing the practicability and efficacy of our algorithm.

(v) An open source code implementation of our algorithm, for reproducing our results and future research [61].

## 3.1 Novelty

$f$-**SVD factorization.** In this work, we suggest a novel factorization technique of an input dataset with respect to a specific loss function $f$, we call it the $f$-SVD factorization. Roughly speaking, the heart of the $f$-SVD factorization lies in finding a diagonal matrix $D \in [0, \infty)^{d \times d}$ and an orthogonal matrix $V \in \mathbb{R}^{d \times d}$ such that the total loss $\sum_{p \in P} f(p, x)$ for any query $x \in \mathbb{R}^d$ can be bounded from above by $\sqrt{d} \left\| DV^T x \right\|_2$ and from below by $\left\| DV^T x \right\|_2$. In some sense, this can be thought of as a $\left( 1 - 1/\sqrt{d} \right)$-coreset (or a sketch) since it approximates the total loss for any query in $\mathbb{R}^d$ up to a multiplicative factor of $\left( 1 - 1/\sqrt{d} \right)$. In order to obtain such factorization, we forge a link between the Löwner ellipsoid [36] and the properties of near-convex functions; see Fig. 1 for a detailed illustrative explanation, Definition 4 and Lemma 16 for the formal details.

Note that *SVD* factorization is a special case of $f$-SVD due to that fact that *SVD* handles functions of the form $\sqrt{\sum_{p \in P} \left| p^T x \right|^2}$ and attempts to achieve the same purpose. The $f$-SVD factorization is a generalization of the *well-conditioned bases* of [21].

**From $f$-SVD to sensitivity bounds.** With the lower bound on the total loss that is guaranteed by the $f$-SVD, we show how to bound the sensitivity of each point in the dataset. On the other hand, the upper bound on the total loss provided by the $f$-SVD factorization, helps us in bounding the total sensitivity. Having this being said, we use the $f$-SVD factorization to suggest a sensitivity bounding framework for a set of points with respect to any near-convex function $f \in \mathcal{F}$; see Lemma 5.

## 4 Preliminaries

**Notations.** For integers $n, d \geq 2$, we denote by $0_d$ the origin of $\mathbb{R}^d$, and by $[n]$ the set $\{1, \cdots, n\}$. The set $\mathbb{R}^{n \times d}$ denotes the union over every $n \times d$ real matrix, and $I_d \in \mathbb{R}^{d \times d}$ denotes the identity matrix. We say that a matrix $A \in \mathbb{R}^{d \times d}$ is *orthogonal* if and only if $A^T A = AA^T = I_d$. Finally, throughout the paper, vectors are addressed as column vectors, and $\Vdash : \mathbb{R}^d \to 1$ is a weight function.

In what follows, we provide formally the notion of $\varepsilon$-coreset in our context.

**Definition 2** ($\varepsilon$-coreset). *Let $P \subseteq \mathbb{R}^d$ be a set of $n$ points, $f : P \times \mathbb{R}^d \to [0, \infty)$ be a near-convex function, and let $\varepsilon \in (0, 1)$. An $\varepsilon$-coreset for $P$ with respect to $f$, is a pair $(S, v)$ where $S \subseteq P$, $v : S \to (0, \infty)$ is a weight function, such that for every $x \in \mathbb{R}^d$, $\left| 1 - \frac{\sum_{q \in S} v(q) f(q, x)}{\sum_{p \in P} f(p, x)} \right| \leq \varepsilon$.*

The following theorem formally describes how to construct an $\varepsilon$-coreset via the sensitivity framework.

**Theorem 3** (Restatement of Theorem 5.5 in [7]). *Let $P \subseteq \mathbb{R}^d$ be a set of $n$ points, and let $f : P \times \mathbb{R}^d \to [0, \infty)$ be a loss function. For every $p \in P$ define the* sensitivity of $p$ as $\sup_{x \in \mathbb{R}^d} \frac{f(p, x)}{\sum_{q \in P} f(q, x)}$, *where the sup is over every $x \in \mathbb{R}^d$ such that the denominator is non-zero. Let $s : P \to [0, 1]$ be*

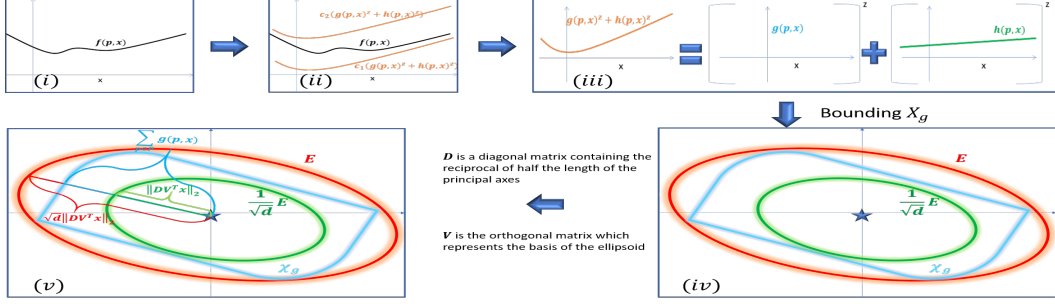

Figure 1: **How to compute $f$-SVD**: (i) Given a set $P \subseteq \mathbb{R}^2$, and a function $f : P \times \mathbb{R}^2 \to [0, \infty)$, (ii) find a function which can bound $f(p, \cdot) \times \mathbb{R}^2 \to [0, \infty)$ from above and below for every $p \in P$, (iii) decompose this function into two functions $g$ and $h$ where for every $p, q \in P$ and $x \in \mathbb{R}^2$, $g(p, \cdot)$ is a convex function (e.g., $g(p, x) = |p^T x|^4$), and $h(p, x) \approx h(q, x)$ (e.g., $h(p, x) = \|x\|_1 + 10$), here $z = 1$. (iv) Since $g$ is convex, we find the Löwner ellipsoid $E$ which contains $\mathcal{X}_g = \{x \in \mathbb{R}^2 | \sum_{p \in P} g(p, x) \leq 1\}$, and the contracted ellipsoid $1/\sqrt{d}E$ is inscribed in $\mathcal{X}_g$. Following this, we compute a diagonal matrix $D \in [0, \infty)^{2 \times 2}$ and an orthogonal matrix $V \in \mathbb{R}^{2 \times 2}$ such that $E = \{x \in \mathbb{R}^2 | \|DV^T x\|_2 \leq 1\}$. (v) By properties of the Löwener ellipsoid, we show that the total loss of $g$ (cyan line) for any query $x \in \mathbb{R}^2$ is in the range $[\|DV^T x\|_2, \sqrt{d} \|DV^T x\|_2]$ (green and red lines). When combined with the bounds on $f$, we obtain an upper bound on the sensitivity of each point in $P$ and on the total sensitivity.

*a function such that $s(p)$ is an upper bound on the sensitivity of $p$. Let $t = \sum_{p \in P} s(p)$ and $d'$ be the VC dimension of the quadruple $(P, \mathbb{1}, \mathbb{R}^d, f)$; see Definition 15. Let $c \geq 1$ be a sufficiently large constant, $\varepsilon, \delta \in (0, 1)$, and let $S$ be a random sample of $|S| \geq \frac{ct}{\varepsilon^2} \left( d' \log t + \log \frac{1}{\delta} \right)$ i.i.d points from $P$, such that every $p \in P$ is sampled with probability $s(p)/t$. Let $v(p) = \frac{t}{s(p)|S|}$ for every $p \in S$. Then, with probability at least $1 - \delta$, $(S, v)$ is an $\varepsilon$-coreset for $P$ with respect to $f$.*

## 5 Coreset for near-convex loss functions

For brevity purposes, proofs of the technical results have been omitted from this manuscript; we refer the reader to the supplementary material for the proofs. In addition, for simplicity of notation, we assume that the weight of each point in the input set is 1, while in the supplementary material, we handle the general case where each point may have any nonnegative weight. We also discuss generalized versions of Definition 1 and Definition 4.

### 5.1 Bounding the sensitivity

The following provides the generalization of the well-conditioned basis of [21], which will be used to bound the sensitivities.

**Definition 4** ($f$-SVD of $P$). *Let $P \subseteq \mathbb{R}^d$ be a set of $n$ points, $f \in \mathcal{F}$ be a near-convex loss function (see Definition 1), and let $g, h, c_1, z$ be defined as in the context of Definition 1 with respect to $f$. Let $D \in [0, \infty)^{d \times d}$ be a diagonal matrix, and let $V \in \mathbb{R}^{d \times d}$ be an orthogonal matrix, such that for every $x \in \mathbb{R}^d$, $c_1 \left( \left( \|DV^T x\|_2 \right)^z + \sum_{p \in P} h(p, x)^z \right) \leq \sum_{p \in P} f(p, x)$, and let $\alpha \in \Theta\left(\sqrt{d}\right)$ such that for every $x \in \mathbb{R}^d$, $\sum_{p \in P} g(p, x)^{\max\{1, z\}} \leq \left( \alpha \|DV^T x\|_2 \right)^{\max\{1, z\}}$. Define $U : P \to \mathbb{R}^d$ such that $U(p) = (VD)^{-1} p$ for every $p \in P$. The tuple $(U, D, V)$ is the $f$-SVD of $P$.*

**Note** that (i) such factorization exists for any set of points $P$ and any near-convex loss function $f : P \times \mathbb{R}^d \to [0, \infty)$ satisfying Definition 1, and (ii) the matrix $VD$ is invertible due to the fact that $D$ is of full rank which is a result of Property (iv) of Definition 4. Both (i)-(ii) hold by using Löwner ellipsoid; see Fig. 1 for intuitive explanation, and Lemma 16 at the Supplementary Material for formal proof.

In what follows, we proceed to bound the sensitivity of each point and the total sensitivity, with respect to a loss function $f \in \mathcal{F}$. This is by using the $f$-SVD of $P$.

**Lemma 5.** *Let $P \subseteq \mathbb{R}^d$ be a set of $n$ points, and let $f \in \mathcal{F}$ be a near-convex loss function as in Definition 1. Let $g, h, c_1, c_2, z$ be defined as in the context of Definition 1 with respect to $f$, $(U, D, V)$*

be the $f$-SVD of $P$, and let $\alpha \in \Theta\left(\sqrt{d}\right)$ which satisfies the conditions in Definition 4. Suppose that there exists a set $\{v_j\}_{j=1}^{O(d)} \subseteq \mathbb{R}^d$ of $O(d)$ unit vectors and $c > 0$, such that for every unit vector $y \in \mathbb{R}^d$ and $p \in P$, $g\left(p, (DV^T)^{-1}y\right)^z \leq c \sum_{j=1}^{O(d)} g\left(p, (DV^T)^{-1}v_j\right)^z$. Then, for every $p \in P$, the sensitivity of $p$ is bounded by $s(p) \leq \frac{2c_2}{c_1 n} + \frac{cc_2}{c_1} \sum_{j=1}^{O(d)} \left(g\left(p, \left(DV^T\right)^{-1} v_j\right)\right)^z$, and the total sensitivity is bounded by $\sum_{p \in P} s(p) \in \frac{2c_2}{c_1} + \frac{cc_2}{c_1} \max\left\{n^{1-z}, 1\right\} \alpha^z O(d)$.

**The existence** of the set $\{v_j\}_{j=1}^{O(d)}$ is discussed in details at the supplementary material at Section D.

## 5.2 The coreset construction

Algorithm 1 receives as input, a set $P$ of $n$ points in $\mathbb{R}^d$, a loss function $f \in \mathcal{F}$ (see Definition 1), and a sample size $m > 0$. As Theorem 6 states, if the sample size $m$ is sufficiently large, then Algorithm 1 outputs a pair $(S, v)$ that is with high probability, an $\varepsilon$-coreset for $P$ with respect to $f$.

First, we set $d'$ to be VC dimension of the quadruple $\left(P, \mathbb{1}, \mathbb{R}^d, f\right)$; See Definition 15. The crux of our algorithm lies in generating the importance sampling distribution via efficiently computing upper bound on the sensitivity of each point (Lines 5–7). To do so, we compute the $f$-SVD of $P$ at Lines 3–4, and we use it to bound the sensitivity of each $p \in P$ as stated in Lemma 5; see Line 6. Now we have all the needed ingredients to use Theorem 3 in order to obtain an $\varepsilon$-coreset, i.e., we sample i.i.d $m$ points from $P$ based on their sensitivity bounds (see Line 9), and assign a new weight for every sampled point at Line 10.

---

**Algorithm 1:** CORESET$(P, f, m)$

---

**Input:**     A set $P \subseteq \mathbb{R}^d$ of $n$ points, a near-convex loss function
              $f : P \times \mathbb{R}^d \to [0, \infty)$, and a sample size $m \geq 1$.
**Output:**    A pair $(S, v)$ that satisfies Theorem 6.

1 Set $d' :=$ the VC dimension of quadruple $\left(P, \mathbb{1}, \mathbb{R}^d, f\right)$ // See Definition 15
2 Set $g$ and $\{z, c_1, c_2\}$ to be a function and a set of real positive numbers respectively, satisfying
   Property (i) and (ii) of Definition 1 with respect to $f$
3 Set $c > 0$ and $\{v_1, \cdots, v_d\}$ to be positive scalar and a set of $d$ unit vectors in $\mathbb{R}^d$ respectively
   satisfying Lemma 5
4 Set $(U, D, V)$ to be the $f$-SVD of $(P, w)$ // See Definition 1
5 **for** *every $p \in P$* **do**
6 $\quad$ Set $s(p) := \frac{cc_2}{c_1} \sum_{j=1}^{d} g\left(p, \left(DV^T\right)^{-1} v_j\right)^z + \frac{2c_2}{c_1 n}$
   $\quad$ // the bound of the sensitivity of $p$ as in Lemma 5
7 Set $t := \sum_{p \in P} s(p)$
8 Set $\tilde{c} \geq 1$ to be a sufficiently large constant // Can be determined from Theorem 6
9 Pick an i.i.d sample $S$ of $m$ points from $P$, where each $p \in P$ is sampled with probability $\frac{s(p)}{t}$.
10 set $v : \mathbb{R}^d \to [0, \infty]$ to be a weight function such that for every $q \in S$, $v(q) = \frac{t}{s(q) \cdot m}$.
11 **return** $(S, v)$

---

**Theorem 6.** *Let $P \subseteq \mathbb{R}^d$ be set of $n$ points, and $f \in \mathcal{F}$ be a near-convex function. Let $R, r > 0$ be a pair of positive scalars as in Definition 1 with respect to $f$, and let $c, c_1, c_2, \alpha$ be defined as in the context of Lemma 5 with respect to $f$. Let $\varepsilon, \delta \in (0, 1)$ be an error parameter and a probability of failure respectively, and let $d'$ be the VC dimension of the triplet $\left(P, f, \mathbb{R}^d\right)$. Let $t = \frac{2c_2}{c_1} + \frac{cc_2}{c_1} \max\left\{n^{1-z}, 1\right\} \alpha^z d$, $m \in O\left(\frac{t}{\varepsilon^2}\left(d' \log(t) + \log\left(\frac{1}{\delta}\right)\right)\right)$, and let $(S, v)$ be the output of a call to CORESET$(P, f, m)$. Then, (i) with probability at least $1 - \delta$, $(S, v)$ is an $\varepsilon$-coreset of size $m$ for $P$ with respect to $f$; see Definition 2. (ii) The overall time for constructing $(S, v)$ is bounded by $O\left(T(n, d)d^4 \log\left(\frac{R}{r}\right)\right)$, where $T(n, d)$ is a bound on the time it takes to compute a gradient of $\sum_{p \in P} f(p, x)$ with respect to any query $x \in \mathbb{R}^d$.*

**Poly-logarithmic coreset size.** We provide an analysis that shows how to obtain a coreset of size poly-logarithmic in the input size $n$; see Algorithm 2 and Lemma 17 at the Supplementary Material.

# 6 Applications

In what follows, we provide various applications for our framework, .e.g, SVM, Logistic Regression, $\ell_z$ for $z \in (0,1)$, outlier resistant functions (similar to Tukey in behavior). For additional problems supported by our framework, we refer the reader to Section G at the Supplementary Material.

Table 1: **Results**: The table below presents the coreset size and the time needed for constructing it with respect to a specific set of problems, where the input is a set of $n$ points in $\mathbb{R}^d$ denoted by $P$. In the table, $\mathbf{nnz}(P)$ denotes the total number of nonzero entries in the set $P$, $\tilde{C}$ denotes the ratio between the number of positive and negative labeled points (in practice, it's a constant number), $\boldsymbol{\lambda} = \sqrt{n}$ is the given regularization parameter for the problems, $\boldsymbol{\gamma} \geq 1$ is defined as in Corollary 10, $\boldsymbol{\varepsilon}$ is the error parameter, and $\boldsymbol{\delta}$ is the probability of failure.

| Problem type | Coreset's size | Construction time[1] |
|---|---|---|
| Logistic regression | $O\left(\frac{d\sqrt{n}}{\varepsilon^2}\left(d\log(d\sqrt{n}) + \log\left(\frac{1}{\delta}\right)\right)\right)$ | $O\left(nd^2\right)$ |
| $\ell_z$-Regression for $z \in (0,1)$ | $O\left(\frac{n^{1-z}d^{\frac{z}{2}+1}}{\varepsilon^2}\left(d\log\left(n^{1-z}d^{\frac{z}{2}+1}\right)\right) + \log\left(\frac{1}{\delta}\right)\right)$ | $O\left(\mathrm{nnz}(P)\log n + d^{O(1)}\right)$ |
| SVM | $O\left(\frac{d\sqrt{n}+\frac{\tilde{C}^2+1}{\tilde{C}}}{\varepsilon^2}\left(d\log\left(d\sqrt{n}+\frac{\tilde{C}^2+1}{\tilde{C}}\right) + \log\left(\frac{1}{\delta}\right)\right)\right)$ | $O\left(nd^2\right)$ |
| Restricted $\ell_z$-regression | $O\left(\frac{\gamma d^{2+\left|\frac{1}{2}-\frac{1}{z}\right|}}{\varepsilon^2}\left(d\log\left(\gamma d^{2+\left|\frac{1}{2}-\frac{1}{z}\right|}\right) + \log\left(\frac{1}{\delta}\right)\right)\right)$ | $O\left(\mathrm{nnz}(P)\log n + d^{O(1)}\right)$ |

**Corollary 7** (Logistic Regression). *Let $P \subseteq \mathbb{R}^d$ be a set of $n$ points such that for every $p \in P$, $\|p\|_2 \leq 1$, $y : P \to \{-1, 1\}$ be a labeling function, $\lambda \geq 1$ be a regularization parameter such that for every $p \in P$, $x \in \mathbb{R}^d$ and $b \in \mathbb{R}$, $f_{\mathrm{LOG}}\left(p, \begin{bmatrix} x \\ b \end{bmatrix}\right) = \frac{1}{\lambda}\ln\left(1 + e^{p^T x + y(p)\cdot b}\right) + \frac{1}{2n}\|x\|_2^2$.*

*Let $\varepsilon, \delta \in (0,1)$ be an error parameter and a probability of failure respectively, $m \in O\left(\frac{dn}{\lambda\varepsilon^2}\left(d\log\left(\frac{dn}{\lambda}\right) + \log\left(\frac{1}{\delta}\right)\right)\right)$, and let $(S, v)$ be the output of a call to $\mathrm{CORESET}(P, f_{\mathrm{LOG}}, m)$. Then, with probability at least $1 - \delta$, $(S, v)$ is an $\varepsilon$-coreset (of size $m$) for $P$ with respect to $f_{\mathrm{LOG}}$.*

**Corollary 8** ($\ell_z$-Regression where $z \in (0,1)$). *Let $P \subseteq \mathbb{R}^d$ be a set of $n$ points, $z \in (0,1)$ and let $f_{\mathrm{NC}\ell_z} : P \times \mathbb{R}^d$ be a loss function such that for every $x \in \mathbb{R}^d$, and $p \in P$, $f_{\mathrm{NC}\ell_z}(p, x) = \left|p^T x\right|^z$.*

*Let $\varepsilon, \delta \in (0,1)$, $m \in O\left(\frac{n^{1-z}d^{\frac{z}{2}+1}}{\varepsilon^2}\left(d\log\left(n^{1-z}d^{\frac{z}{2}+1}\right) + \log\left(\frac{1}{\delta}\right)\right)\right)$, and let $(S, v)$ be the output of a call to $\mathrm{CORESET}(P, f_{\mathrm{NC}\ell_z}, m)$. Then, with probability at least $1 - \delta$, $(S, v)$ is an $\varepsilon$-coreset (of size $m$) for $P$ with respect to $f_{\mathrm{NC}\ell_z}$.*

We now show how our framework can be used to compute an $\varepsilon$-coreset for some query spaces where the involved loss functions are not from the family $\mathcal{F}$. The coreset construction algorithms are hidden in the constructive proofs of the following corollaries.

**Corollary 9** (Support Vector Machines). *Let $P \subseteq \mathbb{R}^d$ be a set of $n$ points such that for every $p \in P$, $\|p\| \leq 1$. Let $y : P \to \{1, -1\}$ be a labelling function, $\lambda \geq 1$ be a regularization parameter such that for every $p \in P$, $x \in \mathbb{R}^d$, and $b \in \mathbb{R}$, $f_{\mathrm{SVM}}\left(p, \begin{bmatrix} x \\ b \end{bmatrix}\right) = \lambda\max\left\{0, 1 - \left(p^T x + y(p)\cdot b\right)\right\} + \frac{1}{2n}\|x\|_2^2$. Let $P_+ = \{p | p \in P, y(p) = 1\}$, $P_- = P \setminus P_+$, $\tilde{C} = \frac{|P_+|}{|P_-|}$.*

*Then, there exists an algorithm that gets the set $P$ as an input, and returns a pair $(S, v)$, such that (i) with probability at least $1 - \delta$, $(S, v)$ is an $\varepsilon$-coreset for $P$ with respect to $f_{\mathrm{SVM}}$, and (ii) the size of the coreset is $|S| \in O\left(\frac{1}{\varepsilon^2}\left(\frac{dn}{\lambda} + \frac{\tilde{C}^2+1}{\tilde{C}}\right)\left(d\log\left(\frac{dn}{\lambda} + \frac{\tilde{C}^2+1}{\tilde{C}}\right) + \log\frac{1}{\delta}\right)\right)$.*

**Corollary 10** (Outlier resistant functions). *Let $P \subseteq \mathbb{R}^d$ be a set of $n$ points, and let $f_{\mathrm{RES}\ell_z} : P \times \mathbb{R}^d \to [0, \infty)$ be loss function such that for every $x \in \mathbb{R}^d$, and $p \in P$, $f_{\mathrm{RES}\ell_z}(p, x) = \min\left\{\left|p^T x\right|, \|x\|_z\right\}$.*

*Then, there exists an algorithm that gets the set $P$ as an in input, and returns a pair $(S, v)$, such that (i) with probability at least $1 - \delta$, $(S, v)$ is an $\varepsilon$-coreset for $P$ with respect to $f_{\mathrm{RES}\ell_z}$, and (ii) the size of the coreset is $O\left(\frac{\gamma d^{2+\left|\frac{1}{2}-\frac{1}{z}\right|}}{\varepsilon^2}\left(d\log\left(\gamma d^{2+\left|\frac{1}{2}-\frac{1}{z}\right|}\right) + \log\left(\frac{1}{\delta}\right)\right)\right)$, where $\gamma$ is defined in the proof.*

# 7 Experimental Results

In what follows we evaluate our coreset against uniform sampling on real-world datasets, with respect to the SVM problem, Logistic regression problem and $\ell_z$-regression problem for $z \in (0, 1)$. Additional details of our setup can be found at Section H of the Supplementary Material.

**Software/Hardware.** Our algorithms were implemented in Python 3.6 [63] using "Numpy" [48], "Scipy" [64] and "Scikit-learn" [49]. Tests were performed on 2.59GHz i7-6500U (2 cores total) machine with 16GB RAM.

**Datasets.** The following datasets were used for our experiments mostly from UCI machine learning repository [22]: (i) **HTRU** [22] — $17,898$ radio emissions of the Pulsar star each consisting of $9$ features. (ii) **Skin** [22] — $245,057$ random samples of R,G,B from face images consisting of $4$ dimensions. (iii) **Cod-rna** [62] — consists of $59,535$ samples, $8$ features, which has two classes (i.e. labels), describing RNAs. (iv) **Web** dataset [9] – $49,749$ web pages records where each record is consists of $300$ features. (v) **3D spatial networks** [22] – 3D road network with highly accurate elevation information (+-20cm) from Denmark used in eco-routing and fuel/Co2-estimation routing algorithms consisting of $434,874$ records where each record has $4$ features.

**Evaluation against uniform sampling.** At Fig. 2a–2f, we have chosen 20 sample sizes, starting from 50 till 500, at Figures 2g–2h, we have chosen 20 sample sizes starting from 4000 till $16,000$. At each sample size, we generate two coresets, where the first is using uniform sampling and the latter is using Algorithm 1. For each coreset $(S, v)$, we find $x^* \in \arg\min_{x \in \mathbb{R}^d} \sum_{p \in S} v(p) f(p, x)$, and the approximation error $\varepsilon$ is set to be $(\sum_{p \in P} f(p, x^*))/(\min_{x \in \mathbb{R}^d} \sum_{p \in P} f(p, x)) - 1$. The results were averaged across 40 trials, while the shaded regions correspond to the standard deviation.

**Evaluation against prior work.** We can not have a fair comparison between our coreset to prior coresets for Logistic regression[47, 56] due to the fact that our formulation of the problem is different. As for support vector machines, we compared our efficacy against [60], the same way that we have compared against uniform sampling. Although not in all cases our approach outperforms [60] in terms of relative error (i.e., $\varepsilon$), our approach is much faster than that of [60]; see Figure 3.

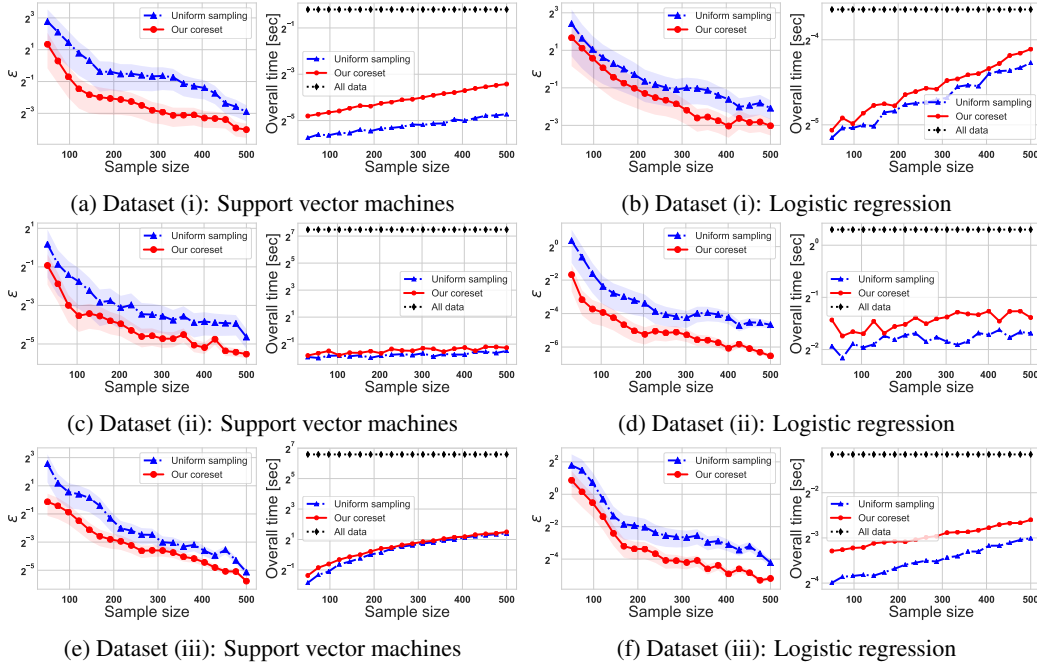

(a) Dataset (i): Support vector machines      (b) Dataset (i): Logistic regression

(c) Dataset (ii): Support vector machines      (d) Dataset (ii): Logistic regression

(e) Dataset (iii): Support vector machines      (f) Dataset (iii): Logistic regression

Figure 2: Experimental results

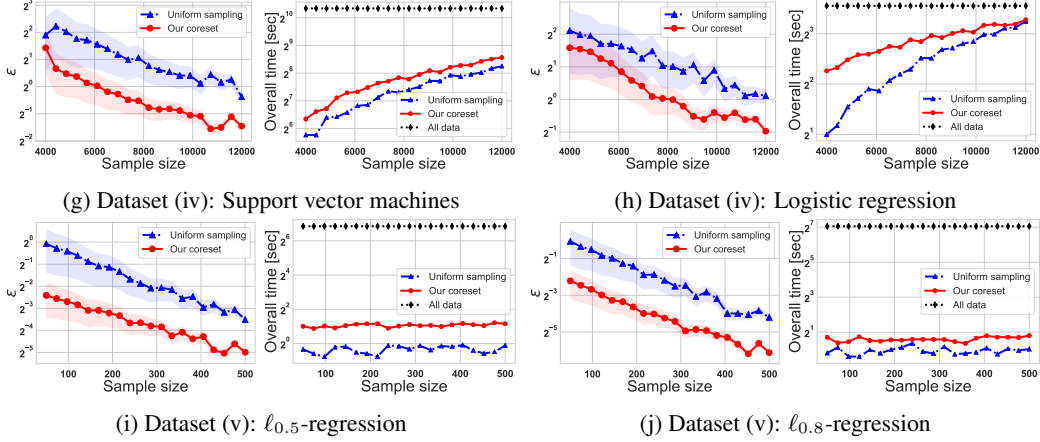

(g) Dataset (iv): Support vector machines      (h) Dataset (iv): Logistic regression

(i) Dataset (v): $\ell_{0.5}$-regression      (j) Dataset (v): $\ell_{0.8}$-regression

Figure 2: Experimental results

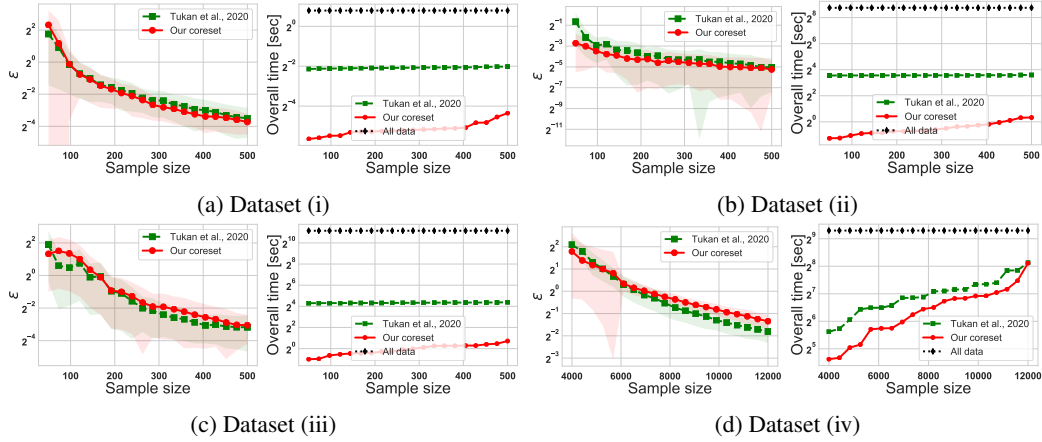

(a) Dataset (i)      (b) Dataset (ii)

(c) Dataset (iii)      (d) Dataset (iv)

Figure 3: Comparison against prior work in the context of SVMs

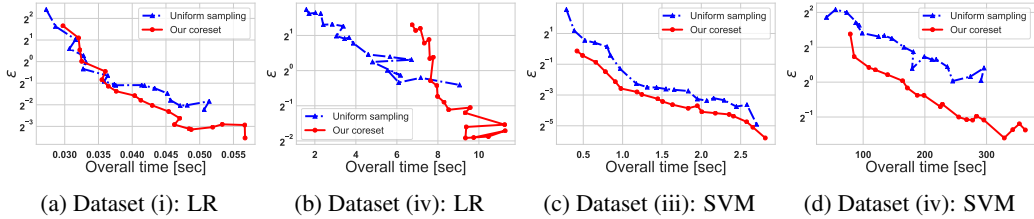

(a) Dataset (i): LR    (b) Dataset (iv): LR    (c) Dataset (iii): SVM    (d) Dataset (iv): SVM

Figure 4: Relative error as a function of the total running time. Here, LR stands for Logistic regression.

## 8   Conclusions and open problems

In this paper, we have provided what we call the $f$-SVD of $P$ with respect a given near-convex loss function $f \in \mathcal{F}$, as well as sensitivity bounding framework using the $f$-SVD. What interests us is to draw back forcing $f$ to have a centrally symmetric level set as well as embedding the center of the Löwner ellipsoid into the sensitivity bound. This is crucial step for generalizing the framework towards a much broader family of functions, e.g., loglog-Lipschitz functions [29]. We are aware that for $\ell_z$-regression problems where $z \geq 1$, Lewis weights have been used by [17] and are considered to be the state of the art coreset for these problems. We aim to generalize the applicability of Lewis weights and other sketching techniques towards different functions, and as far as we know, we consider the above issues to be open problems.

## Broader Impact

Our work provides a strong theoretical result, where we have suggested a generic framework for bounding the sensitivity with respect to broad family of functions. Practically, this family imposes widely used applications such as *SVM*, *Logistic regression*, $\ell_z$-Regression and more.

Although, Broader Impact discussion is not directly applicable, our work can be used to accelerate many known machine learning solvers under various settings such as distributed, streaming, etc.

## Footnotes

[1]Problems which are reduced to $\ell_z$-regression problems for any $z \geq 1$, are easier to be dealt with in term of coreset construction time due to the existence of randomized algoritm of computing the Löwner ellipsoid by [15]; see Section H for detailed description.

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
