[Supplementary Material]

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

## A  Generalization of our tools

We first define the term *query space* which will aid us in simplifying the proofs as well as the corresponding theorems.

**Definition 11** (Query space). *Let $P$ be a set of $n \geq 1$ points in $\mathbb{R}^d$, $w : P \to [0, \infty)$ be a non-negative weight function, and let $f : P \times \mathbb{R}^d \to [0, \infty)$ denote a loss function. The tuple $(P, w, \mathbb{R}^d, f)$ is called a query space.*

Our paper relies on using known theorems associated with convex loss functions to prove our technical results. Thus, for completeness we give a formal definition of a convex loss functions as follows.

**Definition 12** (Convex loss function). *Let $P \subseteq \mathbb{R}^d$ be a set of $n$ points, and let $f : P \times \mathbb{R}^d \to [0, \infty)$ be a loss function. We say that $f$ is a convex loss function if for every $p \in P$, $f(p, \cdot) : \mathbb{R}^d \to [0, \infty)$ is a convex function i.e., for every $\theta \in [0, 1]$ and every $x, y \in \mathbb{R}^d$*

$$f(p, \theta x + (1 - \theta)y) \leq \theta f(p, x) + (1 - \theta)f(p, y).$$

Below, we present a straightforward generalization of the properties in Definition 1, is applied to grasp much more variety of functions, by taking the weights into account and not setting them to 1 for every point in the input set of points as well as other properties.

**Definition 13** (Generalization of Definition 1). *Let $(P, w, \mathbb{R}^d, f)$ be a query space, where $f : P \times \mathbb{R}^d \to [0, \infty)$ is a loss function. We call $f$ a near-convex loss function if there exists a convex loss function $g : P \times \mathbb{R}^d \to [0, \infty)$, a function $h : P \times \mathbb{R}^d \to [0, \infty)$ and a scalar $z > 0$ that satisfies:*

    *(i) There exist $c_1, c_2 \in (0, \infty)$ such that for every $p \in P$, and $x \in \mathbb{R}^d$,*

$$c_1 \left( g(p, x)^z + h(p, x)^z \right) \leq f(p, x) \leq c_2 \left( g(p, x)^z + h(p, x)^z \right).$$

    *(ii) There exist $c_3, c_4 \in (0, \infty)$ such that for every $p \in P$, $x \in \mathbb{R}^d$ and $b \in (0, \infty)$,*

$$c_3 b g(p, x) \leq g(p, bx) \leq c_4 b g(p, x).$$

    *(iii) There exists $c_5 \in (0, \infty)$ such that for every $p \in P$ and $x \in \mathbb{R}^d$,*

$$\frac{w(p)h(p, x)^z}{\sum_{q \in P} w(q)h(p, x)^z} \leq \frac{c_5 w(p)}{\sum_{q \in P} w(q)}.$$

    *(iv) The set $\mathcal{X}_g = \left\{ x \in \mathbb{R}^d \middle| \sum_{p \in P} w(p)^{\max\left\{1, \frac{1}{z}\right\}} g(p, x)^{\max\{1, z\}} \leq 1 \right\}$ is centrally symmetric, i.e., for every $x \in \mathcal{X}_g$ we have $-x \in \mathcal{X}_g$, and there exist $R, r \in (0, \infty)$ such that $B(0_d, r) \subset \mathcal{X}_g \subset B(0_d, R)$, where $B(x, y)$ denotes a ball of radius $y > 0$, centered at $x \in \mathbb{R}^d$.*

*We denote by $\mathcal{F}$ the union of all functions $f$ with the above properties.*

Due to such changes, we also give a generalization towards the definition of $f$-SVD, as in what follows.

**Definition 14** (Generalization of Definition 4). *Let $(P, w, \mathbb{R}^d, f)$ be a query space, such that $f \in \mathcal{F}$, and let $g, h, c_1, c_2, c_3, c_4, z$ be defined as in the context of Definition 1 with respect to $f$. Let $D, V \in \mathbb{R}^{d \times d}$ be a diagonal matrix and an orthogonal matrix respectively, and let $\alpha \in \Theta\left(\sqrt{d}\right)$ such that for every $x \in \mathbb{R}^d$,*

$$c_1 \left( \left(c_3 \left\| DV^T x \right\|_2 \right)^z + \sum_{p \in P} w(p)h(p, x)^z \right) \leq \sum_{p \in P} w(p)f(p, x),$$

*and*

$$\sum_{p \in P} w(p)^{\max\left\{1, \frac{1}{z}\right\}} g(p, x)^{\max\{1, z\}} \leq \left( c_4 \alpha \left\| DV^T x \right\|_2 \right)^{\max\{1, z\}}.$$

*Let $U : P \to \mathbb{R}^d$ such that $U(p) = (VD)^{-1}p$ for every $p \in P$. The tuple $(U, D, V)$ is the $f$-SVD of $(P, w)$.*

# B   VC dimension

**Definition 15** (VC-dimension [7]). *For a query space $(P, w, \mathbb{R}^d, f)$ and $r \in [0, \infty)$, we define*

$$\text{ranges}(x, r) = \{p \in P \mid w(p)f(p, x) \le r\},$$

*for every $x \in \mathbb{R}^d$ and $r \ge 0$. The dimension of $(P, w, \mathbb{R}^d, f)$ is the size $|S|$ of the largest subset $S \subset P$ such that*

$$\left| \left\{ S \cap \text{ranges}(x, r) \mid x \in \mathbb{R}^d, r \ge 0 \right\} \right| = 2^{|S|},$$

*where $|A|$ denotes the number of points in $A$ for every $A \subseteq \mathbb{R}^d$.*

# C   Existence of $f$-SVD factorization

**Lemma 16.** *Let $(P, w, \mathbb{R}^d, f)$ be a query space, such that $f \in \mathcal{F}$. Let $g, h, c_1, c_2, c_3, c_4, z$ be defined as in the context of Definition 13 with respect to $f$, $\alpha \in \Theta\left(\sqrt{d}\right)$ and let $\beta = \max\{1, z\}$. Then there exists a diagonal matrix $D \in \mathbb{R}^{d \times d}$ and an orthogonal matrix $V \in \mathbb{R}^{d \times d}$ such that for every $x \in \mathbb{R}^d$,*

$$\sum_{p \in P} w(p)^{\max\left\{1, \frac{1}{z}\right\}} g(p, x)^\beta \le \left(c_4 \alpha \left\| DV^T x \right\|_2 \right)^\beta, \tag{1}$$

*and*

$$c_1 \left( \left(c_3 \left\| DV^T x \right\|_2 \right)^z + \sum_{p \in P} w(p)h(p, x)^z \right) \le \sum_{p \in P} w(p)f(p, x). \tag{2}$$

*Proof.* We prove Lemma 16 using Löwner ellipsoid; See [36].

**Using Löwner ellipsoid.** Let $\mathcal{X}_g = \left\{ x \in \mathbb{R}^d \middle| \left( \sum_{p \in P} w(p)^{\max\left\{1, \frac{1}{z}\right\}} g(p, x)^\beta \right)^{\frac{1}{\beta}} \le 1 \right\}$. Since $f \in F$ (see Definition 13), and $g$ is convex, we have that (i) $\mathcal{X}_g$ is a convex set, and (ii) $\mathcal{X}_g$ is centrally symmetric. Then by Theorem III of [36], there exists an ellipsoid $E$, known as the Löwner ellipsoid that is centered at the origin $0_d$, such that

$$\frac{1}{\sqrt{d}} E \subseteq X_g \subseteq E, \tag{3}$$

where $\frac{1}{\sqrt{d}} E$ denotes the set $\left\{ \frac{1}{\sqrt{d}} x \mid x \in E \right\}$.

By combining Property (iv) of Definition 13 with (3), there exists $r \in (0, \infty)$ such that $B(0_d, r) \subseteq \mathcal{X}_g \subseteq E$. Since $B(0_d, r) \subseteq E$, then $E$ is an ellipsoid where each of its axes has positive length. By that, there exists a diagonal matrix $D \in \mathbb{R}^{d \times d}$ of positive entries and an orthogonal matrix $V \in \mathbb{R}^{d \times d}$ such that, (i) $E = \left\{ y \in \mathbb{R}^d \mid \left\| DV^T y \right\|_2 \le 1 \right\}$, and (ii) $V^T DDV$ is a positive semi-definite matrix.

Put $x \in \mathbb{R}^d$ and now we proceed to derive the bounds.

**Proving** (1). Let $y = \frac{1}{\|DV^T x\|_2} x$. By the definition of $E$ in (i) and the definition of $y$, we have that $y \in E$, and by combining (3) with the assumption that $\alpha \in \Theta\left(\sqrt{d}\right)$ we obtain that $y \in E \subseteq \alpha \mathcal{X}_g$. Then

$$\frac{1}{\alpha} y \in \frac{1}{\alpha} E \subseteq X_g, \tag{4}$$

which consequently leads to

$$\sum_{p \in P} w(p)^{\max\left\{1, \frac{1}{z}\right\}} g\left(p, \frac{1}{\alpha} y\right)^\beta \le 1, \tag{5}$$

where the inequality holds by Property (iv) of Definition 13 with respect to $g$. Hence,

$$\sum_{p\in P} w(p)^{\max\{1,\frac{1}{z}\}} g(p,x)^\beta \le \left(c_4\alpha \left\|DV^Tx\right\|_2\right)^\beta \sum_{p\in P} w(p)^{\max\{1,\frac{1}{z}\}} g\left(p,\frac{x}{\alpha\left\|DV^Tx\right\|_2}\right)^\beta \tag{6}$$
$$\le \left(c_4\alpha \left\|DV^Tx\right\|_2\right)^\beta,$$

where the first inequality is by substituting $b := \alpha\left\|DV^Tx\right\|_2$ and $x := \frac{x}{b}$ in Property (ii) of $g$ (see Definition 13), and the second inequality is by combining the fact that $y = \frac{1}{\left\|DV^Tx\right\|_2}x$ with (5).

**Proving** (2). Let $b' = \alpha\left\|DV^Tx\right\|_2$. By (5), we get that $\sum_{p\in P} w(p)^{\max\{1,\frac{1}{z}\}} g\left(p,\frac{1}{b'}x\right)^\beta \ge 1$. In addition, Property (iv) of Definition 13 states that every vector in $\mathbb{R}^d$ of norm $r$ is inside $\mathcal{X}_g$. Thus, there exists $b \ge b'$ such that $\sum_{p\in P} w(p)^{\max\{1,\frac{1}{z}\}} g\left(p,\frac{1}{b}x\right)^\beta = 1$.

By (3), we have that $\left\|DV^Tx\right\|_2 = b\left\|DV^Tz\right\|_2 \le b$ where $z = \frac{1}{b}x$. Hence, by plugging $x := z$ and $b := b$ in Property (ii) of Definition 13, we obtain that

$$\sum_{p\in P} w(p)^{\max\{1,\frac{1}{z}\}} g\left(p,x\right)^\beta \ge c_3 b \sum_{p\in P} w(p)^{\max\{1,\frac{1}{z}\}} g\left(p,z\right)^\beta = (c_3 b)^\beta \ge \left(c_3\left\|DV^Tx\right\|_2\right)^\beta. \tag{7}$$

By combining Property (i) of Definition 13, with (6) and (7), Lemma 16 holds. $\qquad\square$

# D The intuition behind the existence of the set $\{v_j\}_{j=1}^{O(d)}$

Let $(P,w,\mathbb{R}^d,f)$ be a query space (see Definition 11) such that $f \in \mathcal{F}$ as in Definition 13. Let $g,c_4,z$ be defined as in the context of Definition 13 with respect to $f$. Since by Definition 13, the level set of $g$ is bounded and is contained in a ball of radius $R$, then it holds that Löwner ellipsoid of the level set of $g$ is contained in the ball $B\left(0_d,\sqrt{d}R\right)$, using similar arguments to those at (3). Then there exist a point $\tilde{c}$ on the ball $B\left(0_d,R\right)$ such that on the ball $B\left(\tilde{c},2d\max\{R,1/R\}\right)$ there exist a set of $d+1$ vectors $\{\tilde{v}_i\}_{i=1}^{d+1}$ each of norm $2dR$ such that any unit vector $x \in \mathbb{R}^d$ is a convex combination of them. For instance, the set $\{\tilde{v}_i\}_{i=1}^{d+1}$ can be computed by finding a $d$-simplex which inscribes the unit ball $B\left(0_d,1\right)$.

Let $A \in \mathbb{R}^{d\times d}$ and observe that by using Jensen's inequality, for any unit vector $x \in \mathbb{R}^d$ and $p \in P$, we have

$$g(p,Ax)^z \le \left(\sum_{i=1}^{d+1} g\left(p,A\tilde{v}_i\right)\right)^z \le d^z \sum_{i=1}^{d+1} g\left(p,A\tilde{v}_i\right)^z \le \left(2c_4 d^2 R\right)^z \sum_{i=1}^{d+1} g\left(p,Av_i\right)^z,$$

where $v_i = \frac{1}{2dR}\tilde{v}_i$ for every $i \in [d+1]$.

Thus the assumption in Lemma 20 holds.

**Note that** this holds in the general case of $g$ however in the applications that we handled, since $g$ is also homogeneous function, then only $d$ vectors suffices for satisfying this assumption in Lemma 20; see Section F.

# E Extension towards Streaming and distributed settings

Algorithm 1 can be easily extended towards streaming and distributed settings as presented at Algorithm 2. At the beginning, the data arrives in a streaming fashion, e.g. in batches, where our coreset scheme (see Algorithm 1) is applied on each of these batches. When we have two $\varepsilon$-coresets in memory, we merge them and an $\varepsilon$-coreset is constructed upon their merge. This procedure is done until (i) there is no points left in the stream and (ii) there is exactly one coreset left in memory.

Algorithm 2 begins with initializing the batches to an empty sets as well as setting the height of the tree to 1; see lines 1–2. In what follows, for each $2l$ of streamed points, we generate an $\varepsilon$-coreset on

**Algorithm 2:** STREAMING-CORESET$(P, w, f, l, \varepsilon, \delta)$

---

**Input:** A set $P \subseteq \mathbb{R}^d$ of $n$ points, a weight function $w : \mathbb{R}^d \to [0, \infty)$, a leaf size $\ell > 0$
a convex loss function function $f : \mathbb{R}^d \to [0, \infty)$, an error parameter $\varepsilon \in (0, 1)$,
and probability $\delta \in (0, 1)$.

**Output:** A pair $(S, v)$ which is an $(h)$-coreset for $(P, \mathbb{R}^d, X, f)$,
with probability of at least $1 - \delta h$.

1 $B_i \leftarrow \emptyset$ for every $1 \leq i \leq \infty$
2 $h \leftarrow 1$
3 **for** *each set $Q$ of consecutive $2\ell$ points from $P$* **do**
4    $(T, v) := \text{CORESET}(Q, w, f, \frac{\varepsilon}{2 \log n}, \frac{\delta}{2 \log n})$
5    $j \leftarrow 1$
6    $B_j := B_j \cup (T, v)$
7    **for** *each $j \leq h$* **do**
8       **while** $|B_j| \geq 2$ **do**
9          $(T_1, v_1), (T_2, v_2) :=$ pop first pair of consecutive items in $B_j$
10         For every $p \in T_1 \cup T_2$, set $v'(p) := \begin{cases} v_1(p) & p \in T_1, \\ v_2(p) & \text{Otherwise} \end{cases}$
11         $(T, v) := \text{CORESET}(T_1 \cup T_2, v', f, \frac{\varepsilon}{2 \log n}, \frac{\delta}{2 \log n})$
12         $B_{j+1} := B_{j+1} \cup (T, v)$
13         $h := \max\{h, j + 1\}$
14 $(S, v) := B_h$
15 **return** $(S, v)$

---

this set as presented at lines 3–6. Lines 7–13 depict the core of the merge-and-reduce tree, which is the binary tree building fashion from the leaves (the incoming batches) towards the root of the tree. Finally, we return the root of the tree as shown at lines 14–15. For much broader and detailed explanation regarding the merge-and-reduce tree, we refer the reader towards [7].

### E.1 From sublinear to poly-logarithmic coreset size

**Lemma 17** (Variant of Lemma 4, [60])**.** *Let $P \subseteq \mathbb{R}^d$ be a set of $n$ points, and let $f \in \mathcal{F}$ be a near-convex loss function. Let $\varepsilon \in \left[\frac{1}{\log n}, \frac{1}{2}\right]$, $\delta \in \left[\frac{1}{\log n}, 1\right)$ and let $t$ denote the total sensitivity from Lemma 5. Suppose that there exists some $\beta \in (0.1, 0.8)$ such that $t \in \Theta(n^\beta)$ and let $\ell \geq 2^{\frac{\beta}{1-\beta}}$. Let $(S, v)$ be the output of a call to STREAMING-CORESET$(P, w, f, \ell, \varepsilon, \delta)$. Then $(S, v)$ is an $\varepsilon$-coreset of size*

$$|S| \in (\log n)^{O(1)}.$$

*Proof.* First we note that using Theorem 6 on each node in the merge-and-reduce tree, would attain that the root of the tree, i.e., $(S, v)$ attains that for every $w$

$$(1 - \varepsilon)^{\log n} \sum_{p \in P} w(p) f(p, x) \leq \sum_{p \in S} v(p) f(p, x) \leq (1 + \varepsilon)^{\log n} \sum_{p \in P} w(p) f(p, x),$$

with probability at least $(1 - \delta)^{\log n}$.

We observe by the properties of the natural number $e$,

$$(1 + \varepsilon)^{\log n} = \left(1 + \frac{\varepsilon \log n}{\log n}\right)^{\log n} \leq e^{\varepsilon \log n},$$

which when replacing $\varepsilon$ with $\varepsilon' = \frac{\varepsilon}{2 \log n}$ in the above inequality as done at Lines 4 and 9 of Algorithm 2, we obtain that

$$(1 + \varepsilon')^{\log n} \leq e^{\frac{\varepsilon}{2}} \leq 1 + \varepsilon, \tag{8}$$

where the second inequality holds since $\varepsilon \in [\frac{1}{\log n}, \frac{1}{2}]$.

As for the lower bound, observe that

$$(1 - \varepsilon)^{\log n} \geq 1 - \varepsilon \log n,$$

where the inequality holds since $\varepsilon \in [\frac{1}{\log n}, \frac{1}{2}]$.

Hence,

$$(1 - \varepsilon')^{\log n} \geq 1 - \varepsilon' \log n = 1 - \frac{\varepsilon}{2} \geq 1 - \varepsilon.$$

Similar arguments holds also for the failure probability $\delta$. What is left for us to do is setting the leaf size which will attain us an $\varepsilon$-coreset of size poly-logarithmic in $n$ (the number of points in $P$).

Let $\ell \in (0, \infty)$ be the size of a leaf in the merge-and-reduce tree. We observe that a coreset of size poly-logarithmic in $n$, can be achieved by solving the inequality

$$\frac{2\ell}{2} \geq (2\ell)^{\beta},$$

which is invoked when ascending from any two leafs and their parent node at the merge-and-reduce tree.

Rearranging the inequality, we yield that

$$\ell^{1-\beta} \geq 2^{\beta}.$$

Since $\ell \in (0, \infty)$, any $\ell \geq \sqrt[1-\beta]{2^{\beta}}$ would be sufficient for the inequality to hold. What is left for us to do, is to show that when ascending through the merge-and-reduce tree from the leaves towards the root, each parent node can't be more than half of the merge of it's children (recall that the merge-and-reduce tree is built in a binary tree fashion, as depicted at Algorithm 2).

Thus, we need to show that,

$$2^{\sum_{j=1}^{i} \beta^j} \cdot \ell^{\beta^i} \leq \frac{2^{\sum_{k=0}^{i-1} \beta^k} \cdot \ell^{\beta^{i-1}}}{2} = 2^{\sum_{k=1}^{i-1} \beta^k} \cdot \ell^{\beta^{i-1}},$$

holds, for any $i \in [\lceil \log n \rceil]$ where $\log n$ is the height of the tree. Note that the left most term is the parent node's size and the right most term represents half the size of both parent's children nodes.

In addition, for $i = 1$, the inequality above represents each node which is a parent of leaves. Thus, we observe that for every $i \geq 1$, the inequality represents ascending from node which is a root of a sub-tree of height $i - 1$ to it's parent in the merge-and-reduce tree.

By simplifying the inequality, we obtain the same inequality which only addressed the leaves. Hence, by using any $\ell \geq 2^{\frac{\beta}{1-\beta}}$ as a leaf size in the merge and reduce tree, we obtain an $\varepsilon$-coreset of size poly-logarithmic in $n$. $\qquad\square$

# F  Proofs for the Main Theorems

Throughout this section, we will present generalized versions of the lemmata and theorems that are presented at Section 5 and Section 6.

## F.1  Generalization of Lemma 5

**Lemma 18** (Equivalence of norms, [51]). *Let $a, b > 0$ such that $a \leq b$. Then for every $x \in \mathbb{R}^d$,*

$$\|x\|_b \leq \|x\|_a \leq d^{\frac{1}{a} - \frac{1}{b}} \|x\|_b.$$

**Claim 19.** *[Result of Hölder's Inequality] Let $\{a_i\}_{i=1}^{n}$ be a set of $n$ non-negative numbers, $z \in (0, 1)$ be a real number. Then*

$$\sum_{i=1}^{n} |a_i|^z \leq n^{1-z} \left( \sum_{i=1}^{n} |a_i| \right)^z.$$

*Proof.* Let $z' = \frac{1}{z}$ and for every $i \in [n]$, let $\hat{a}_i = |a_i|^z$. Let $e \in [1]^n$. We have

$$\sum_{i=1}^{n} |a_i|^z = \sum_{i=1}^{n} \hat{a}_i \leq \|e\|_{\frac{1}{1-z}} \left( \sum_{i=1}^{n} \hat{a}_i^{z'} \right)^{1/z'} = n^{1-z} \left( \sum_{i=1}^{n} |a_i| \right)^z,$$

where the first and last equalities are by definition of $\hat{a}_i$, and the inequality is by Hölder's inequality. $\square$

**Lemma 20.** *Let $(P, w, \mathbb{R}^d, f)$ be a query space (see Definition 11) such that $f \in \mathcal{F}$ as in Definition 13. Let $g, h, c_1, c_2, c_3, c_4, c_5, z$ be defined as in the context of Definition 13 with respect to $f$, $(U, D, V)$ be the $f$-SVD of $(P, w)$, and let $\alpha \in \Theta\left(\sqrt{d}\right)$ which satisfies the conditions in Definition 4. Suppose that there exists a set of $O(d)$ unit vectors $\{v_j\}_{j=1}^{O(d)}$ and $c \in (0, \infty)$, such that for every unit vector $y \in \mathbb{R}^d$ and $p \in P$,*

$$g(p, (DV^T)^{-1}y)^z \leq c \sum_{j=1}^{O(d)} g(p, (DV^T)^{-1}v_j)^z.$$

*Then, for every $p \in P$, the sensitivity of $p$ with respect to the query space $(P, w, \mathbb{R}^d, f)$ is bounded by*

$$s(p) \leq \left( \frac{2c_2 c_5 w(p)}{c_1 \sum_{q \in P} w(q)} \right)^z + \frac{cc_2}{c_1 c_3^{2z}} \sum_{j=1}^{O(d)} w(p) \left( g\left( p, (DV^T)^{-1}v_j \right) \right)^z,$$

*and the total sensitivity is bounded by*

$$\sum_{p \in P} s(p) \in \frac{c_2 c_5}{c_1} + \frac{cc_2 c_4^z}{c_1 c_3^{2z}} \max\left\{ n^{1-z}, 1 \right\} \alpha^z O(d).$$

*Proof.* Let $n$ denote the number of points in $P$. Put $p \in P$, $x \in \mathbb{R}^d$ such that $\sum_{q \in P} w(q)f(q, x) > 0$, and let $y = \frac{1}{\|DV^T x\|_2} DV^T x$. We observe that,

$$\frac{f(p, x)}{\sum\limits_{q \in P} w(q)f(q, x)} \leq \frac{f(p, x)}{c_1 \left( (c_3 \|DV^T x\|_2)^z + \sum\limits_{q \in P} w(q)h(q, x)^z \right)} \tag{9}$$

$$\leq \frac{c_2 g(p, x)^z + c_2 h(p, x)^z}{c_1 (c_3 \|DV^T x\|_2)^z + c_1 \sum\limits_{q \in P} w(q)h(q, x)^z} \tag{10}$$

$$\leq \frac{c_2 g(p, x)^z}{c_1 (c_3 \|DV^T x\|_2)^z} + \frac{c_2 h(p, x)^z}{c_1 \sum\limits_{q \in P} h(q, x)^z}, \tag{11}$$

where (9) holds by Lemma 16, (10) holds by Property (i) of Definition 1 with respect to $f$, and the last inequality follows from plugging $a_1 := c_2 g(p, x)$, $r_1 := c_2 h(p, x)$, $a_2 := c_1 c_3 \|DV^T x\|_2$ and $r_2 := c_3 \sum_{q \in P} h(q, x)$ into Claim 21.

Note that when $h(q, z) = 0$ for every $q \in P$ and $z \in \mathbb{R}^d$, then we obtain from (10), that the rightmost term of (11) is zero.

We also have,

$$\frac{1}{\|DV^T x\|_2^z} g(p, x)^z \leq \frac{1}{c_3^z} g\left( p, \frac{x}{\|DV^T x\|_2} \right)^z \tag{12}$$

$$= \frac{1}{c_3^z} g\left( p, (DV^T)^{-1} y \right)^z \tag{13}$$

$$\leq \frac{c}{c_3^z} \sum_{j=1}^{d} g\left( p, (DV^T)^{-1} v_j \right)^z, \tag{14}$$

where (12) follows from substituting $b := \frac{1}{\|DV^T x\|_2}$ and $x := \frac{x}{b}$ in Property (ii) of $f$ (see Definition 1), (13) holds since $x = (DV^T)^{-1}(DV^T)x$, and (14) is by the assumption of Lemma 5.

By combining (9)–(14) with Property (iii) of $f$, the sensitivity of $p$ is bounded by

$$s(p) \leq \frac{c_2 c_5 w(p)}{c_1 \sum_{q \in P} w(q)} + \frac{cc_2}{c_1 c_3^{2z}} \sum_{j=1}^{d} w(p) g\left(p, \left(DV^T\right)^{-1} v_j\right)^z. \tag{15}$$

As for the total sensitivity, we first observe that if $z \in (0,1)$

$$\sum_{q \in P} \sum_{j=1}^{O(d)} w(q) g\left(q, \left(DV^T\right)^{-1} v_j\right)^z = \sum_{j=1}^{O(d)} \sum_{q \in P} w(q) g\left(q, \left(DV^T\right)^{-1} v_j\right)^z \tag{16}$$

$$= \sum_{j=1}^{O(d)} \sum_{q \in P} \left(w(q)^{\frac{1}{z}} g\left(q, \left(DV^T\right)^{-1} v_j\right)\right)^z \tag{17}$$

$$\leq n^{1-z} \sum_{j=1}^{O(d)} \left(\sum_{q \in P} w(q)^{\frac{1}{z}} g\left(q, \left(DV^T\right)^{-1} v_j\right)\right)^z \tag{18}$$

$$\leq n^{1-z} \left(c_4 \alpha\right)^z \sum_{j=1}^{O(d)} \left\|DV^T \left(DV^T\right)^{-1} v_j\right\|_2^z \tag{19}$$

$$= n^{1-z} \left(c_4 \alpha\right)^z \sum_{j=1}^{O(d)} \|v_j\|_2 \tag{20}$$

$$= n^{1-z} \left(c_4 \alpha\right)^z O(d), \tag{21}$$

where (16) holds by the independency between the summation over $q \in P$ and summation over $j \in [d]$, (17) holds since the weights are non-negative by definition, (18) holds by plugging $z := z$, $n := n$, $a_i := w(q)^{\frac{1}{z}} g\left(q, \left(DV^T\right)^{-1} v_j\right)$ for every $i \in [n]$ into Claim 19 where $q$ denotes the $i$th point in $P$, (19) holds by Lemma 16, (20) follows since $DV^T \left(DV^T\right)^{-1} = I_d$, and finally (21) follows from the assumption of Lemma 5.

Similarly for the case of $z \geq 1$,

$$\sum_{q \in P} \sum_{j=1}^{O(d)} w(q) g\left(q, \left(DV^T\right)^{-1} v_j\right)^z \in \left(c_4 \alpha\right)^z O(d). \tag{22}$$

Hence, Lemma 5 holds as

$$\sum_{p \in P} s(p) \leq \sum_{p \in P} \frac{c_2 c_5 w(p)}{c_1 \sum_{q \in P} w(q)} + \sum_{p \in P} \frac{cc_2}{c_1 c_3^{2z}} \sum_{j=1}^{O(d)} w(p) g\left(p, \left(DV^T\right)^{-1} v_j\right)^z$$

$$\in \frac{c_2 c_5}{c_1} + \frac{cc_2 c_4^z}{c_1 c_3^{2z}} \max\left\{n^{1-z}, 1\right\} \alpha^z O(d),$$

where the first inequality holds by (15), and the second inequality holds by combining (16)–(22). □

## F.2 Proof of Theorem 6

**Theorem 6.** *Let $P \subseteq \mathbb{R}^d$ be set of $n$ points, and $f \in \mathcal{F}$ be a near-convex function. Let $R, r > 0$ be a pair of positive scalars as in Definition 1 with respect to $f$, and let $c, c_1, c_2, \alpha$ be defined as in the context of Lemma 5 with respect to $f$. Let $\varepsilon, \delta \in (0,1)$ be an error parameter and a probability of failure respectively, and let $d'$ be the VC dimension of the triplet $\left(P, f, \mathbb{R}^d\right)$. Let $t = \frac{2c_2}{c_1} + \frac{cc_2}{c_1} \max\left\{n^{1-z}, 1\right\} \alpha^z d$, $m \in O\left(\frac{t}{\varepsilon^2}\left(d' \log(t) + \log\left(\frac{1}{\delta}\right)\right)\right)$, and let $(S, v)$ be the output of a call to CORESET$(P, f, m)$. Then, (i) with probability at least $1 - \delta$, $(S, v)$ is an $\varepsilon$-coreset of size*

$m$ for $P$ with respect to $f$; see Definition 2. (ii) The overall time for constructing $(S, v)$ is bounded by $O\left(T(n, d)d^4 \log\left(\frac{R}{r}\right)\right)$, where $T(n, d)$ is a bound on the time it takes to compute a gradient of $\sum_{p \in P} f(p, x)$ with respect to any query $x \in \mathbb{R}^d$.

*Proof.* In algorithm 1, we first compute the sensitivity bounds $s(p)$ for every $p \in P$ with respect to the query space $(P, w, \mathbb{R}^d, f)$. This is done based on Lemma 5; See Line 6. We then sample a sufficiently large number of points based on those sensitivity bound as Theorem 3 states; See Line 9. Hence, By plugging $P, w, f, \mathbb{R}^d, \varepsilon, \delta$ and $s(p)$ for every $p \in P$ into Theorem 3, we obtain that with probability at least $1 - \delta$, $(S, v)$ is an $\varepsilon$-coreset (see Definition 2) of size $|S| \in O\left(\frac{t}{\varepsilon^2}\left(d' \log t + \log\left(\frac{1}{\delta}\right)\right)\right)$.

The overall time is dominated by computing the $f$-SVD of $(P, w)$, i.e., $(U, D, V)$ at Line 4 of Algorithm 1. This is done by computing the Löwner ellipsoid, as explained in the proof of Lemma 16.

The computation of the Löwner ellipsoid, requires a separation oracle, where we use the gradient of $g$ as a candidate, similarly to [10]. We refer to [41] for more details on the computation of Löwner ellipsoid. $\square$

### F.3 Proof of Corollary 7

**Claim 21.** *Let $a_1, r_1, a_2, r_2 \in [0, \infty)$ such that $a_2, r_2 > 0$. Then,*

$$\frac{a_1 + r_1}{a_2 + r_2} \leq \frac{a_1}{a_2} + \frac{r_1}{r_2}.$$

*Proof.* Observe that,

$$\frac{a_1 + r_1}{a_2 + r_2} = \frac{a_1}{a_2 + r_2} + \frac{r_1}{a_2 + r_2} \leq \frac{a_1}{a_2} + \frac{r_1}{r_2},$$

where the inequality holds since $a_2, r_2 > 0$ and $a_1, r_1 \geq 0$. $\square$

**Claim 22.** *Let $N \geq 2$. For every $i \in [N]$, let $a_i \geq 0$ and $b_i > 0$. Then,*

$$\frac{\max\{a_1, a_2, \cdots, a_N\}}{\max\{b_1, b_2, \cdots, b_N\}} \leq \max\left\{\frac{a_1}{b_1}, \frac{a_2}{b_2}, \cdots, \frac{a_N}{b_N}\right\}.$$

*Proof.* Let $\hat{i} \in \arg\max_{i \in [N]} a_i$ and let $\hat{j} \in \arg\max_{i \in [N]} b_i$. Then,

$$\frac{\max\{a_1, a_2, \cdots, a_N\}}{\max\{b_1, b_2, \cdots, b_N\}} = \frac{a_{\hat{i}}}{b_{\hat{j}}} \leq \frac{a_{\hat{i}}}{b_{\hat{i}}} \leq \max\left\{\frac{a_1}{b_1}, \frac{a_2}{b_2}, \cdots, \frac{a_N}{b_N}\right\},$$

where the first inequality holds by the definition of $b_{\hat{j}}$.

$\square$

**Claim 23.** *For every $z, b \in \mathbb{R}$,*

$$\ln\left(1 + e^{z+b}\right) \leq 2\ln\left(1 + e^{z^2} e^b\right).$$

*Proof.* Put $b \in \mathbb{R}$, and note that for every $z \in \mathbb{R}$, we have

$$\ln(2) + z^2 - z \geq 0,$$

by rearranging the above, we get that

$$\ln(2) + z^2 \geq z.$$

Applying the exponentiation operation on both sides with respect to the natural number $e$ as the base, yields

$$2e^{z^2} \geq e^z,$$

and since $e^{z^2+b} > 0$,

$$e^z \leq e^{z^2}\left(2 + e^{z^2+b}\right).$$

By multiplying each side by $e^b$ and adding 1, we obtain that

$$1 + e^{z+b} \leq 1 + 2e^{z^2+b} + e^{z^2+2b}.$$

Applying the logarithm function on both sides of the inequality above proves Claim 23 as

$$\ln\left(1 + e^{z+b}\right) \leq 2\ln\left(1 + e^{z^2}e^b\right).$$

$\square$

**Lemma 24** (Bernoulli's inequality, [39]). *Let $x \geq -1$ be a real number and let $r \in [0,1]$ be a positive real number. Then,*

$$(1+x)^r \leq 1 + rx$$

**Lemma 25.** *Let $N > 1$, $c \in [1, N]$ and let $p \in \mathbb{R}^d$ such that $\|p\|_2 \leq 1$. Then for every $(x, b) \in \mathbb{R}^d \times \mathbb{R}$,*

*(i) $\frac{1}{c}\ln\left(1 + e^{p^T x + b}\right) + \frac{1}{2N}\|x\|_2^2 \leq \frac{4}{c}(p^T x)^2 + 4\max\left\{\frac{1}{c}\ln\left(1 + e^b\right), \frac{1}{2N}\|x\|_2^2\right\}$,*

*(ii) and $\frac{1}{c}\ln\left(1 + e^{p^T x + b}\right) + \frac{1}{2N}\|x\|_2^2 \geq \frac{c}{8N}\left(\frac{1}{c}(p^T x)^2 + \max\left\{\frac{1}{c}\ln\left(1 + e^b\right), \frac{1}{2N}\|x\|_2^2\right\}\right)$.*

*Proof.* Put $(x, b) \in \mathbb{R}^d \times \mathbb{R}$. We now proceed to prove Lemma 25.

**Proof of Claim (i).** By plugging $z := p^T x$ and $b := b$ into Claim 23, we obtain that

$$\ln\left(1 + e^{p^T x + b}\right) \leq 2\ln\left(1 + e^{(p^T x)^2 + b}\right) \leq 2\ln\left(e^{(p^T x)^2}\left(1 + e^b\right)\right) = 2\left(p^T x\right)^2 + 2\ln\left(1 + e^b\right),$$
(23)

where the second inequality holds since $e^{(p^T x)^2} \geq 1$, and the equality follows from properties of the logarithm function.

Thus, Claim (i) holds since

$$\frac{1}{c}\ln\left(1 + e^{p^T x + b}\right) + \frac{1}{2N}\|x\|_2^2 \leq \frac{2}{c}\left(p^T x\right)^2 + \frac{2}{c}\ln\left(1 + e^b\right) + \frac{1}{2N}\|x\|_2^2$$

$$\leq \frac{4}{c}\left(p^T x\right)^2 + 4\max\left\{\frac{1}{c}\ln\left(1 + e^b\right), \frac{1}{2N}\|x\|_2^2\right\},$$

where the first inequality is by (23), the second inequality holds by properties of the *max* operator.

**Proof of Claim (ii).** We start by noting that since $\|p\| \leq 1$, we have that

$$\|x\|_2 \geq \left|p^T x\right|,$$
(24)

which consequently leads to

$$\frac{1}{c}\ln\left(1 + e^{p^T x + b}\right) \geq \frac{1}{c}\ln\left(1 + e^{-\|x\|_2 + b}\right) \geq \frac{1}{2N}\ln\left(1 + e^{-\|x\|_2 + b}\right),$$
(25)

where the second inequality holds since $c \leq N$.

We show that

$$\ln\left(1 + e^{-\|x\|_2 + b}\right) + \|x\|^2 \geq \frac{1}{2}\ln\left(1 + e^b\right),$$
(26)

holds for every $x \in \mathbb{R}^d$ and $b \in \mathbb{R}$. In order to to that, we first define the function $q : \mathbb{R} \to (0, \infty)$ such that for every $r \in \mathbb{R}$, $q(r) = \ln\left(1 + e^{-|r| + b}\right) + r^2$.

Let $W$ denotes the *Lambert W function* ( see [19]). Minimizing $q(r)$ over $r \in \mathbb{R}$, requires computing the derivative of $q(r)$ with respect to $r$, and setting it to zero. We observe that when setting the derivative to zero we obtain that $r^* \in [-W(1), W(1)]$, i.e., the left term of (26) attains its minimal value at some $x^* \in \mathbb{R}^d$ such that $\|x^*\|_2 \in [0, W(1)]$.

Observe that for every $x \in \mathbb{R}^d$

$$\ln\left(1 + e^{-\|x\|_2 + b}\right) + \|x\|_2^2 \geq \ln\left(1 + e^{-\|x^*\|_2 + b}\right) + \|x^*\|_2^2 \geq \ln\left(1 + e^{-\|x^*\|_2 + b}\right) \geq \ln\left(1 + e^{-W(1) + b}\right),$$

where the first inequality holds by the definition of $x^*$, the second inequality holds since $\|x^*\|_2^2 \geq 0$, and the last inequality follows from the observation that $\|x^*\| \in [0, W(1)]$.

Since $e^{-W(1)} \in (0, 1)$, we have that

$$\ln\left(1 + e^{-W(1)+b}\right) \geq \ln\left(\left(1 + e^b\right)^{e^{-W(1)}}\right) = e^{-W(1)} \ln\left(1 + e^b\right) \geq \frac{1}{2} \ln\left(1 + e^b\right),$$

where the first inequality holds by plugging $r := e^{-W(1)}$ and $x := e^b$ into Lemma 24, the equality holds by properties of the logarithm function, and the last inequality holds since $e^{-W(1)} \geq \frac{1}{2}$.

We also observe that

$$\frac{1}{2N} \|x\|_2^2 \geq \frac{1}{2N} \left|p^T x\right|^2 = \frac{c}{2N} \left(\frac{1}{c} \left|p^T x\right|^2\right), \tag{27}$$

where the first inequality holds by (24), and the equality holds since $\frac{c}{2N} \cdot \frac{1}{c} = \frac{1}{2N}$.

Thus by combining (25), (26), and (27), Claim (ii) holds as

$$\frac{1}{c} \ln\left(1 + e^{p^T x + b}\right) + \frac{1}{2N} \|x\|_2^2 \geq \frac{c}{4N} \left(\frac{1}{2c} \left|p^T x\right|^2 + \max\left\{\frac{1}{2c} \ln\left(1 + e^b\right), \frac{1}{2N} \|x\|_2^2\right\}\right).$$

$\square$

**Lemma 26.** *Let* $(P, w, \mathbb{R}^{d+1}, f_{\mathrm{LOG}})$ *be a query space,* $y : P \to \{1, -1\}$ *be a labelling function,* $\lambda \geq 1$ *be a regularization parameter, such that for every* $p \in P$, $b \in \mathbb{R}$ *and* $x \in \mathbb{R}^d$,

$$f_{\mathrm{LOG}}(p, (x \mid b)) = \frac{1}{2 \sum\limits_{q \in P} w(q)} \|x\|_2^2 + \frac{1}{\lambda} \ln\left(1 + e^{p^T x + y(p)b}\right).$$

*For every* $p \in P$, *let* $P_{y(p)} = \{q \mid q \in P, y(q) = y(p)\}$ *denote the set of points with the same label as the label assigned to* $p$. *Let* $(U, D, V)$ *be the* $f$-*SVD of* $(P, w)$ *with respect to* $f_{\mathrm{LOG}}$. *Then, claims (i) – (ii) hold as follows:*

*(i) for every* $p \in P$, *the sensitivity of* $p$ *with respect to the query space* $(P, w, \mathbb{R}^{d+1}, f_{\mathrm{LOG}})$ *is bounded by*

$$s(p) = \frac{32}{\lambda} \left(\frac{2w(p)}{\sum\limits_{q \in P_{y(p)}} w(q)} + w(p) \|U(p)\|_2^2\right) \sum_{q \in P_{y(p)}} w(q),$$

*(ii) and the total sensitivity is bounded by*

$$\sum_{p \in P} s(p) \leq \frac{32}{\lambda} (2 + d) \sum_{p \in P} w(p).$$

*Proof.* Put $p \in P$ and let $P_{y(p)}$ denote the subset of points from $P$ with same label as $p$, $P_{y(p)} = \{q \mid q \in P, y(q) = y(p)\}$. Observe that for every $q \in P_p$

$$\sup_{(x,b) \in \mathbb{R}^d \times \mathbb{R}} \frac{w(p) f_{\mathrm{LOG}}(p, (x \mid b))}{\sum\limits_{q \in P} w(q) f_{\mathrm{LOG}}(q, (x \mid b))} \leq \sup_{(x,b) \in \mathbb{R}^d \times \mathbb{R}} \frac{w(p) f_{\mathrm{LOG}}(p, (x \mid b))}{\sum\limits_{q \in P_{y(p)}} w(q) f_{\mathrm{LOG}}(q, (x \mid b))},$$

where the inequality holds since $P_{y(p)} \subseteq P$, and $f_{\mathrm{LOG}}(q, (x \mid b)) \geq 0$ for every $q \in P$, and $(x \mid b) \in \mathbb{R}^d \times \mathbb{R}$.

Note the following:

(a) For every $q \in P$, $x \in \mathbb{R}^d$ and $\gamma \geq 0$ we have $\left|q^T \gamma x\right| = \gamma \left|q^T x\right|$.

(b) Since $\left|q^T x\right|$ is convex function, it also holds that $\sum\limits_{q\in P}\left|q^T x\right|^2$ is convex due to the fact that sum of convex functions is also convex,

(c) The level set $\left\{x \,\middle|\, x\in\mathbb{R}^d, \sum\limits_{q\in P} w(q)\left|q^T x\right|^2 \leq 1\right\}$ is convex and is centrally symmetric.

(d) For every $x\in\mathbb{R}^d$, and $b\in\mathbb{R}$ we have that

$$\frac{w(q)\max\left\{\frac{1}{\lambda}\ln\left(1+e^b\right), \frac{1}{2\sum\limits_{q\in P_{y(p)}} w(q)}\|x\|_2^2\right\}}{\sum\limits_{\tilde{q}\in P_{y(p)}} w(\tilde{q})\max\left\{\frac{1}{\lambda}\ln\left(1+e^b\right), \frac{1}{2\sum\limits_{q\in P_{y(p)}} w(q)}\|x\|_2^2\right\}} \leq 2\frac{w(q)}{\sum\limits_{\tilde{q}\in P_{y(p)}} w(\tilde{q})},$$

where the inequality holds by plugging $a_1 := w(q)\frac{1}{\lambda}\ln\left(1+e^b\right)$, $a_2 := \frac{1}{2\sum\limits_{q\in P_{y(p)}} w(q)}\|x\|_2^2$,

$b_1 := \frac{\sum\limits_{\tilde{q}\in P_{y(p)}} w(\tilde{q})}{\lambda}\ln\left(1+e^b\right)$, $b_2 := \frac{1}{2}\|x\|_2^2$ into Claim 22.

Thus, combining (a), (b), (c), (d) and Lemma 25, allows us to plug

- $f(p,(x\mid b)) := f_{\text{LOG}}(p,(x\mid b))$, $g(p,(x\mid b)) := \frac{1}{\sqrt{\lambda}}\left|p^T x\right|$ and $h(p,(x\mid b)) :=$

  $\max\left\{\sqrt{\frac{1}{\lambda}\ln\left(1+e^b\right)}, \sqrt{\frac{1}{2\sum\limits_{q\in P_{y(p)}} w(q)}\|x\|_2^2}\right\}$, for every $p\in P$, $x\in\mathbb{R}^d$, $b\in\mathbb{R}$,

- $\alpha = 1$,

- $c_1 := \frac{\lambda}{8N}$ and $c_2 = 4$,

- $c_i := 1$ for every $i\in[3,4]$

- $c_5 := 2$,

- $z := 2$,

- $v_j := e_j$ for every $j\in[d]$ where $e_j$ denotes the vector with a 1 in the $j$th coordinate and 0's elsewhere,

- and $c := 1$,

into Lemma 5, which yields that $f_{\text{LOG}}\in\mathcal{F}$ and the sensitivity of each point $q\in P_{y(p)}$ is bounded by

$$s(q) = \frac{32}{\lambda}\left(2\frac{w(q)}{\sum\limits_{\tilde{q}\in P_{y(p)}} w(\tilde{q})} + w(q)\sum_{j=1}^d\left|U(q)^T e_j\right|^2\right)\sum_{\tilde{q}\in P_{y(p)}} w(\tilde{q}).$$

Claim (i) now holds since for every $q\in P$,

$$\sum_{j=1}^d\left|U(q)^T e_j\right|^2 = \|U(q)\|_2^2,$$

where the equality follows from definition of $e_j$ for every $j\in[d]$.

As for the total sensitivity, we have by Lemma 5,

$$\sum_{q\in P_{y(p)}} s(q) \leq \frac{32}{\lambda}\left(2\frac{w(q)}{\sum\limits_{\tilde{q}\in P_{y(p)}} w(\tilde{q})} + d\right)\sum_{q\in P_{y(p)}} w(p),$$

and

$$\sum_{q \in P \setminus P_{y(p)}} s(q) \leq \frac{32}{\lambda} \left( 2 \frac{w(q)}{\sum_{\tilde{q} \in P \setminus P_{y(p)}} w(\tilde{q})} + d \right) \sum_{q \in P \setminus P_{y(p)}} w(p).$$

Hence, Claim (ii) holds as

$$\sum_{q \in P} s(q) \leq \frac{32}{\lambda} (2 + d) \sum_{q \in P} w(q).$$

$\square$

**Corollary 7** (Logistic Regression). *Let $P \subseteq \mathbb{R}^d$ be a set of $n$ points such that for every $p \in P$, $\|p\|_2 \leq 1$, $y : P \to \{-1, 1\}$ be a labeling function, $\lambda \geq 1$ be a regularization parameter such that for every $p \in P$, $x \in \mathbb{R}^d$ and $b \in \mathbb{R}$, $f_{\mathrm{LOG}} \left( p, \begin{bmatrix} x \\ b \end{bmatrix} \right) = \frac{1}{\lambda} \ln \left( 1 + e^{p^T x + y(p) \cdot b} \right) + \frac{1}{2n} \|x\|_2^2$.*

*Let $\varepsilon, \delta \in (0, 1)$ be an error parameter and a probability of failure respectively, $m \in O \left( \frac{dn}{\lambda \varepsilon^2} \left( d \log \left( \frac{dn}{\lambda} \right) + \log \left( \frac{1}{\delta} \right) \right) \right)$, and let $(S, v)$ be the output of a call to $\mathrm{CORESET}(P, f_{\mathrm{LOG}}, m)$. Then, with probability at least $1 - \delta$, $(S, v)$ is an $\varepsilon$-coreset (of size $m$) for $P$ with respect to $f_{\mathrm{LOG}}$.*

*Proof.* First, observe that by Lemma 26 the total sensitivity is bounded by $t := \frac{32}{\lambda} (2 + d) \sum_{q \in P} w(p)$. Hence, plugging $s(p)$ for every $p \in P$ from Lemma 26, $t := t$, $\varepsilon := \varepsilon$ and $\delta := \delta$ into Theorem 6, yields that $(S, v)$ is an $\varepsilon$-coreset of size $O \left( \frac{dC}{\lambda \varepsilon^2} \left( d \log \left( \frac{dC}{\lambda} \right) + \log \left( \frac{1}{\delta} \right) \right) \right)$. $\square$

### F.4 Proof of Corollary 8

**Lemma 27.** *Let $(P, w, \mathbb{R}^{d+1}, f_{\mathrm{NC}\ell_z})$ be a query space, such that for every $p \in P$ and $x \in \mathbb{R}^d$,*

$$f_{\mathrm{NC}\ell_z}(p, x) = \left| p^T x \right|^z .$$

*Let $(U, D, V)$ be the $f$-SVD of $(P, w)$ with respect to $f_{\mathrm{NC}\ell_z}$. Then, claims (i) – (ii) hold as follows:*

(i) *for every $p \in P$, the sensitivity of $p$ with respect to the query space $(P, w, \mathbb{R}^d, f_{\mathrm{NC}\ell_z})$ is bounded by*

$$s(p) = w(p) \|U(p)\|_z^z ,$$

(ii) *and the total sensitivity is bounded by*

$$\sum_{p \in P} s(p) \leq n^{1-z} d^{\frac{z}{2}+1}.$$

*Proof.* Let $g : P \to [0, \infty)$ such that for every $p \in P$ and $x \in \mathbb{R}^d$, $g(p, x) = \left| p^T x \right|$, and for every $i \in [d]$ let $e_i$ denote the vector with 1 in the $i$th coordinate and 0's elsewhere. Observe that:

(a) For every $q \in P$, $x \in \mathbb{R}^d$ and $b \geq 0$ we have $g(p, b \cdot x) = b \cdot g(p, x)$.

(b) Since $g(q, x)$ is a convex function for every $q \in P$, it also holds that $\sum_{q \in P} w(q)^{\frac{1}{z}} g(q, x)$ is convex due to the fact that sum of convex functions is also convex.

(c) The level set $\left\{ x \middle| x \in \mathbb{R}^d, \sum_{q \in P} w(q)^{\frac{1}{z}} g(q, x) \leq 1 \right\}$ is convex and is centrally symmetric.

(d) In addition, for every unit vector $y \in \mathbb{R}^d$

$$\left| p^T y \right|^z \leq \|p\|_2^z \leq \|p\|_z^z = \sum_{i=1}^d \left| p^T e_i \right|^z ,$$

where the first inequality holds by Cauchy's inequality, the second inequality is by Lemma 18, and the equality is by properties of norm.

Hence combining (a), (b), (c) and (d), allows us to plug

- $f(q, x) := f_{\mathrm{NC}\ell_z}(q, x)$, $g(q, x) := |q^T x|$ and $h(q, x) := 0$ for every $q \in P$ and $x \in \mathbb{R}^d$,

- $c_i := 1$ for every $i \in [4]$,

- $c_5 := 0$,

- $\alpha := \sqrt{d}$,

- $v_j := e_j$ for every $j \in [d]$ where $e_j$ denotes the vector with a 1 in the $j$th coordinate and 0's elsewhere,

- $c := 1$, and

- $z := z$

into Lemma 5, which yields that $f_{\mathrm{NC}\ell_z} \in \mathcal{F}$ and the sensitivity of each point $p \in P$ is bounded by by

$$s(p) = \sum_{i=1}^{d} \left| U(p)^T e_i \right|_z^z,$$

and the total sensitivity is bounded by

$$\sum_{q \in P} s(q) \le n^{1-z} d^{\frac{z}{2}+1}.$$

$\square$

**Corollary 8** ($\ell_z$-Regression where $z \in (0,1)$). *Let $P \subseteq \mathbb{R}^d$ be a set of $n$ points, $z \in (0,1)$ and let $f_{\mathrm{NC}\ell_z} : P \times \mathbb{R}^d$ be a loss function such that for every $x \in \mathbb{R}^d$, and $p \in P$, $f_{\mathrm{NC}\ell_z}(p, x) = \left| p^T x \right|^z$.*

*Let $\varepsilon, \delta \in (0,1)$, $m \in O\left( \frac{n^{1-z} d^{\frac{z}{2}+1}}{\varepsilon^2} \left( d \log \left( n^{1-z} d^{\frac{z}{2}+1} \right) + \log \left( \frac{1}{\delta} \right) \right) \right)$, and let $(S, v)$ be the output of a call to $\mathrm{CORESET}\,(P, f_{\mathrm{NC}\ell_z}, m)$. Then, with probability at least $1 - \delta$, $(S, v)$ is an $\varepsilon$-coreset (of size $m$) for $P$ with respect to $f_{\mathrm{NC}\ell_z}$.*

*Proof.* First, observe that by Lemma 27, the total sensitivity is bounded by $t := n^{1-z} d^{\frac{z}{2}+1}$. Plugging $s(p)$ for every $p \in P$ from Lemma 27, $t := t$, $\varepsilon := \varepsilon$ and $\delta := \delta$ into Theorem 6, yields an $\varepsilon$-coreset of size $O\left( \frac{n^{1-z} d^{\frac{z}{2}+1}}{\varepsilon^2} \left( d \log \left( n^{1-z} d^{\frac{z}{2}+1} \right) + \log \left( \frac{1}{\delta} \right) \right) \right)$. $\square$

### F.5 Proof of Corollary 9

**Lemma 28.** *Let $\gamma \in (0,1]$, $N \ge 1$, and let $c \in [1, N]$. Let $p \in \mathbb{R}^d$ such that $\|p\|_2 \le 1$ and let $X = \left\{ (x, b) \in \mathbb{R}^d \times \mathbb{R} \mid \|x\|_2 \ge \gamma, |b| \le 9 \|x\|_2 \right\}$. Then, for every $(x, b) \in X$, claims (i) – (ii) hold as follows:*

(i) $\frac{\|x\|_2^2}{N} + \frac{1}{c} \max \left\{ 0, 1 + p^T x + b \right\} \le \frac{2}{c} \left| p^T x \right|^2 + 2 \max \left\{ \frac{1}{c}, \frac{b}{c}, \frac{\|x\|_2^2}{N} \right\}$,

(ii) $\frac{\|x\|_2^2}{N} + \frac{1}{c} \max \left\{ 0, 1 + p^T x + b \right\} \ge \frac{c\gamma^2}{(1+10\gamma)N} \left( \frac{1}{c} \left| p^T x \right|^2 + \max \left\{ \frac{1}{c}, \frac{b}{c}, \frac{\|x\|_2^2}{N} \right\} \right)$.

*Proof.* Put $(x, b) \in X$.

**Proof of Claim (i).** The proof is by the following case analysis:

1. If $p^T x + b \ge 0$, we have

$$\max \left\{ 0, 1 + p^T x + b \right\} = 1 + p^T x + b \le 2 + 2 \left| p^T x \right|^2 + b \le 2 \left| p^T x \right|^2 + 2 \max \left\{ 1, b \right\},$$

where the equality holds by the assumption of the case, and the first inequality holds since for every $z \in \mathbb{R}$, we have $1 + z \le 2z^2 + 2$.

2. Otherwise,

$$\max\left\{0, 1 + p^T x + b\right\} \le 1 \le 2\max\left\{1, b\right\} \le 2\left|p^T x\right|^2 + 2\max\left\{1, b\right\}$$

where the first inequality holds by the assumption of the case.

By taking both the above cases in mind, Claim (i) holds.

**Proof of Claim (ii).**   Similar to the proof of Claim (i), we use the same case analysis:

1. If $p^T x + b \ge 0$, we observe that

$$
\begin{aligned}
\frac{1}{c}\max\left\{0, 1 + p^T x + b\right\} + \frac{\|x\|_2^2}{N} &= \frac{1}{c}\left(1 + p^T x + b\right) + \frac{\|x\|_2^2}{N} \\
&\ge \frac{c}{2N}\left(\frac{1}{c}\left|p^T x\right|^2 + \max\left\{1, b, \frac{1}{2N}\|x\|_2^2\right\}\right),
\end{aligned}
\tag{28}
$$

where the equality holds by the assumption of this case, and the inequality holds since $\left|p^T x\right|^2 \le \|x\|_2^2$ due to the assumption that $\|p\|_2 \le 1$.

2. Otherwise,

$$\frac{\left|p^T x\right|^2}{c} + \max\left\{\frac{1}{c}, \frac{b}{c}, \frac{\|x\|_2^2}{N}\right\} \le \frac{1}{c} + \frac{10\|x\|_2^2}{\gamma c},\tag{29}$$

where the inequality follows since by the definition of the set $X$, we have that $b \le 9\|x\|_2 \le \frac{9\|x\|_2^2}{\gamma}$.

We also note that,

$$\frac{1}{c}\max\left\{0, 1 + p^T x + b\right\} + \frac{\|x\|_2^2}{N} \ge \frac{\|x\|_2^2}{N},\tag{30}$$

holds since the $\max$ term is non-negative.

Let $l = \frac{c\gamma^2}{N(1 + 10\gamma)}$. Observe that

$$\frac{l}{c}\left(1 + \frac{10\|x\|^2}{\gamma}\right) \le \frac{\|x\|_2^2}{N},\tag{31}$$

since $\|x\|_2 \ge \gamma$.

Hence, we obtain that

$$\frac{1}{c}\max\left\{0, 1 + p^T x + b\right\} + \frac{\|x\|_2^2}{N} \ge \frac{c\gamma^2}{N(1 + 10\gamma)}\left(\frac{\left|p^T x\right|^2}{c} + \max\left\{\frac{1}{c}, \frac{b}{c}, \frac{1}{2N}\|x\|_2^2\right\}\right),$$

where the inequality holds by combining (29), (30) and (31).

Combining both cases proves Claim (ii).

$\square$

**Lemma 29.** *Let* $(P, w, \mathbb{R}^{d+1}, f_{\mathrm{SVM}})$ *be a query space,* $y : P \to \{1, -1\}$ *be a labelling function,* $\lambda \ge 1$ *be a regularization parameter such that for every* $p \in P$, $x \in \mathbb{R}^d$, *and* $b \in \mathbb{R}$,

$$f_{\mathrm{SVM}}(p, (x \mid b)) = \frac{1}{2\sum_{q \in P} w(q)}\|x\|_2^2 + \lambda\max\left\{0, 1 - \left(p^T x + y(p)b\right)\right\}.$$

*For every* $p \in P$, *let* $P_{y(p)} = \{q \mid q \in P, y(q) = y(p)\}$ *denote the set of points with the same label as the label assigned to* $p$.

*Let* $(U, D, V)$ *be the* $f$*-SVD of* $(P, w)$ *with respect to* $f_{\mathrm{SVM}}$. *Then, claims* $(i) - (ii)$ *hold as follows:*

*(i) for every $p \in P$, the sensitivity of $p$ with respect to the query space $(P, w, \mathbb{R}^{d+1}, f_{\text{SVM}})$ is bounded by*

$$s(p) = \max \left\{ \frac{9w(p)}{\sum\limits_{q \in P_{y(p)}} w(q)}, \frac{2w(p)}{\sum\limits_{q \in P \setminus P_{y(p)}} w(q)} \right\} + \frac{13w(p)}{4 \sum\limits_{q \in P_{y(p)}} w(q)}$$

$$+ \frac{125 \sum\limits_{q \in P} w(q)}{4\lambda} \cdot \left( w(q) \left\| U(p) \right\|_2^2 + \frac{w(p)}{\sum\limits_{q \in P} w(q)} \right),$$

*(ii) and the total sensitivity is bounded by*

$$\sum_{p \in P} s(p) \leq 25 + \frac{\sum\limits_{p \in P_{y(p)}} w(p)}{\sum\limits_{q \in P \setminus P_{y(p)}} w(q)} + \frac{\sum\limits_{q \in P \setminus P_{y(p)}} w(p)}{\sum\limits_{p \in P} w(q)} + \frac{125 \sum\limits_{q \in P} w(q)}{4C} \cdot (d+2).$$

*Proof.* Put $p \in P$, let $P_{y(p)}$ denote the subset of points from $P$ with same label as $p$, i.e., $P_{y(p)} = \{q \mid q \in P, y(q) = y(p)\}$, let $\gamma = 0.4$, and let $X = \{(x, b) \mid x \in \mathbb{R}, b \in \mathbb{R}, \|x\|_2 \leq \gamma\}$. We have that

$$\sup_{(x,b) \in \mathbb{R}^d \times \mathbb{R}} \frac{w(p) f_{\text{SVM}}(p, (x \mid b))}{\sum\limits_{q \in P} w(q) f_{\text{SVM}}(q, (x, b))}$$

$$\leq \sup_{(x,b) \in X} \frac{w(p) f_{\text{SVM}}(p, (x \mid b))}{\sum\limits_{q \in P} w(q) f_{\text{SVM}}(q, (x \mid b))} + \sup_{(x,b) \in \mathbb{R}^d \times \mathbb{R} \setminus X} \frac{w(p) f_{\text{SVM}}(p, (x \mid b))}{\sum\limits_{q \in P} w(q) f_{\text{SVM}}(q, (x \mid b))}, \quad (32)$$

**Proof of Claim (i).** By the above inequality, in order to bound the sensitivity of a point $p \in P$, we can bound the term in 32. For that, we first proceed to bound the left hand side of (32).

**Handling queries from $X$.** We observe that for every $q \in P$

$$-\gamma \leq -\gamma \left\| q \right\|_2 \leq -\left\| x \right\|_2 \left\| q \right\|_2 \leq q^T x \leq \left\| x \right\|_2 \left\| q \right\|_2 \leq \gamma \left\| q \right\|_2 \leq \gamma, \quad (33)$$

where the first and last inequalities hold since $\|q\| \leq 1$ for every $q \in P$, the second and fifth inequalities hold since $\|x\|_2 \leq \gamma$, and the third and forth inequalities hold by Cauchy-Schwartz's inequality.

In addition,

$$\sup_{(x,b) \in X} \frac{w(p) f_{\text{SVM}}(p, (x \mid b))}{\sum\limits_{q \in P} w(q) f_{\text{SVM}}(q, (x \mid b))} \leq \sup_{(x,b) \in X} \frac{w(p) \left\| x \right\|_2^2}{\sum\limits_{q \in P} w(q) \left\| x \right\|_2^2} + \sup_{(x,b) \in X} \frac{w(p) \max \left\{ 0, 1 + p^T x + y(p)b \right\}}{\sum\limits_{q \in P} w(q) \max \left\{ 0, 1 + q^T x + y(q)b \right\}},$$

$$(34)$$

where the inequality holds by plugging $a_1 := \frac{w(p)}{2 \sum\limits_{q \in P} w(q)} \left\| x \right\|_2^2$, $r_1 :=$

$w(p) \max \left\{ 0, 1 + p^T x + y(p)b \right\}$, $a_2 := \frac{1}{2} \left\| x \right\|_2^2$ and $r_2 := \sum\limits_{q \in P} w(q) f_{\text{SVM}}(q, (x, b)) - \frac{1}{2} \left\| x \right\|_2$

into Claim 21.

Bounding the rightmost term of (34) requires carefully checking three cases:

(a) If $y(p)b > 0$, then we have

$$\frac{w(p)\max\left\{0, 1 + p^T x + y(p)b\right\}}{\sum\limits_{q \in P} w(q)\max\left\{0, 1 + q^T x + y(q)b\right\}} \leq \frac{w(p)\max\left\{0, 1 + p^T x + y(p)b\right\}}{\sum\limits_{q \in P_{y(p)}} w(q)\max\left\{1 + q^T x + y(q)b\right\}}$$

$$= \frac{w(p)\left(1 + p^T x + y(p)b\right)}{\sum\limits_{q \in P_{y(p)}} w(q)\left(1 + q^T x + y(q)b\right)}$$

$$\leq \frac{w(p)\left(1 + p^T x\right)}{\sum\limits_{q \in P_{y(p)}} w(q)\left(1 + q^T x\right)} + \frac{y(p)w(p)b}{\sum\limits_{q \in P_{y(p)}} y(q)w(q)b},$$

$$(35)$$

where the first inequality holds since $P_{y(p)} \subseteq P$, the equality follows from combining the assumption that $\gamma \in (0,1)$ and (33), and the last inequality holds by combining the fact that $1 + q^T x \geq 0$ for every $q \in P_{y(q)}$, the assumption of the case, and the result of plugging $a_1 := w(p)\left(1 + p^T x\right)$, $r_1 := w(p)y(p)b$, $a_2 := \sum\limits_{q \in P_{y(p)}} w(q)\left(1 + q^T x\right)$ and $r_2 := \sum\limits_{q \in P_{y(p)}} y(q)w(q)b$ into Claim 21.

We also have

$$\frac{w(p)\max\left\{0, 1 + p^T x\right\}}{\sum\limits_{q \in P} w(q)\max\left\{0, 1 + q^T x\right\}} \leq \frac{w(p)\max\left\{0, 1 + \gamma\|p\|_2\right\}}{\sum\limits_{q \in P} w(q)\max\left\{0, 1 - \gamma\|q\|_2\right\}}$$

$$\leq \frac{w(p)\left(1 + \gamma\right)}{\sum\limits_{q \in P} w(q)\left(1 - \gamma\right)},$$

where the first inequality holds by (33), and the second inequality follows from the assumption that for every $q \in P$, $\|q\| \leq 1$.

(b) If $y(p)b \in [-\gamma, 0]$, then

$$\frac{w(p)\max\left\{0, 1 + p^T x + y(p)b\right\}}{\sum\limits_{q \in P} w(q)\max\left\{0, 1 + q^T x + y(q)b\right\}} \leq \frac{w(p)\max\left\{0, 1 + \gamma\|p\|_2 + \gamma\right\}}{\sum\limits_{q \in P_{y(p)}} w(q)\max\left\{0, 1 - \gamma\|q\|_2 - \gamma\right\}} \leq \frac{w(p)\max\left\{0, 1 + 2\gamma\right\}}{\sum\limits_{q \in P_{y(p)}} w(q)\left(1 - 2\gamma\right)},$$

where the first inequality holds since $|y(p)b| \leq \gamma$ and $P_{y(p)} \subseteq P$, and the second inequality holds since $\gamma \in \left(0, \frac{1}{2}\right)$.

(c) Otherwise, we have $-\gamma > y(p)b$, which means that for every $q \in P$ such that $y(q) \neq y(p)$, we have $\gamma < y(q)b$.

Thus,

$$\frac{w(p)\max\left\{0, 1 + p^T x + y(p)b\right\}}{\sum\limits_{q \in P} w(q)\max\left\{0, 1 + q^T x - \gamma\right\}} \leq \frac{w(p)\max\left\{0, 1 + p^T x + y(p)b\right\}}{\sum\limits_{q \in P \setminus P_{y(p)}} w(q)\max\left\{0, 1 + q^T x - \gamma\right\}}$$

$$\leq \frac{w(p)\max\left\{0, 1 + \gamma\|p\|_2 + \gamma\right\}}{\sum\limits_{q \in P \setminus P_{y(p)}} w(q)\max\left\{0, 1 - \|q\|_2\gamma + \gamma\right\}}$$

$$\leq \frac{w(p)}{\sum\limits_{q \in P \setminus P_{y(p)}} w(q)},$$

where the first inequality holds since $P \setminus P_{y(p)} \subseteq P$, the second inequality follows from (33), and the last inequality holds by the assumption that $\|q\|_2 \leq$ for every $q \in P$.

Since $\gamma \in \left(0, \frac{1}{2}\right)$, we have $1 \le \frac{1+\gamma}{1-\gamma} \le \frac{1+2\gamma}{1-2\gamma}$, and by that we get

$$\frac{w(p)\max\{0, 1+\gamma\}}{\sum\limits_{q \in P} w(q)(1-\gamma)} \le \frac{w(p)\max\{0, 1+2\gamma\}}{\sum\limits_{q \in P} w(q)(1-2\gamma)}. \tag{36}$$

Combining the cases above with (36), yields that

$$\sup_{(x,b) \in X} \frac{w(p) f_{\mathrm{SVM}}(p, (x \mid b))}{\sum\limits_{q \in P} w(q) f_{\mathrm{SVM}}(q, (x \mid b))} \le 2\max\left\{\frac{w(p)(1+2\gamma)}{\sum\limits_{q \in P_{y(p)}} w(q)(1-2\gamma)}, \frac{w(p)}{\sum\limits_{q \in P \setminus P_{y(p)}} w(q)}\right\}. \tag{37}$$

**Handling queries from $\mathbb{R}^d \times \mathbb{R} \setminus X$.** Put $(x, b) \in \mathbb{R}^d \times \mathbb{R} \setminus X$, and consider the following case analysis:

(a) If $|b| \le 9\|x\|_2$, then we note the following:

  (A) For every $q \in P$, $x \in \mathbb{R}^d$ and $\beta \ge 0$ we have $|q^T \beta x| = \beta \cdot |q^T x|$.

  (B) Since $|q^T x|$ is a convex function for every $q \in P$, it also holds that $\sum\limits_{q \in P} w(q) |q^T x|^2$ is convex due to the fact that sum of convex functions is also convex.

  (C) The level set $\left\{x \,\middle|\, x \in \mathbb{R}^d, \ \sum\limits_{q \in P} w(q) |q^T x|^2 \le 1\right\}$ is convex and is centrally symmetric.

  (D) In addition, for every unit vector $y \in \mathbb{R}^d$

$$|q^T y|^2 \le \|q\|_2^2 = \sum_{i=1}^{d} |q^T e_i|^2,$$

  where the inequality holds by Cauchy's inequality and the equality holds by properties of norm.

  By combining (A), (B), (C), (D) and the result of substituting $c := \lambda$, $N := 2\sum\limits_{q \in P} w(q)$ and $\gamma := 0.4$ into Lemma 28, we get that we can plug

  - $f(p, (x \mid b)) := f_{\mathrm{SVM}}(p, (x \mid b))$, $g(p, (x \mid b)) := \frac{1}{\lambda}|p^T x|$, and $h(p, (x \mid b)) := \max\left\{\frac{1}{\lambda}, \frac{b}{\lambda}, \frac{\|x\|_2^2}{N}\right\}$
  - $\alpha := 1$,
  - $c_1 := \frac{\lambda\gamma^2}{(1+10\gamma)\sum\limits_{q \in P} w(q)}$ and $c_2 := 2$,
  - $c_i := 1$ for every $i \in [3, 4]$
  - $c_5 := 2$,
  - $v_j := e_j$ for every $j \in [d]$ where $e_j$ denotes the vector with a 1 in the $j$th coordinate and 0's elsewhere,
  - and $c := 1$,

  into Lemma 5, to obtain that $f_{\mathrm{SVM}} \in \mathcal{F}$ with respect to any $x \in \mathbb{R}^d \times \mathbb{R} \setminus X$ and the sensitivity $p$ is bounded by

$$\frac{w(p) f_{\mathrm{SVM}}(p, (x \mid b))}{\sum_{q \in P} w(q) f_{\mathrm{SVM}}(q, (x \mid b))} \le \frac{(1+10\gamma)\sum\limits_{q \in P} w(q)}{\gamma^2 \lambda}\left(\frac{2w(p)}{\sum_{q \in P} w(q)} + \sum_{j=1}^{d} |U(p)^T e_j|^2\right), \tag{38}$$

  with respect to any query in $\mathbb{R}^d \times \mathbb{R} \setminus X$.

(b) If $y(p)b \geq 9\|x\|_2$ then we have that

$$\frac{w(p)f_{\text{SVM}}(p,(x\mid b))}{\sum_{q\in P} w(q)f_{\text{SVM}}(q,(x\mid b))} \leq \frac{w(p)\|x\|_2^2}{\|x\|_2^2 \sum_{q\in P} w(q)} + \frac{w(p)\max\left\{0, 1+p^T x + y(p)b\right\}}{\sum_{q\in P} w(q)\max\left\{0, 1+q^T x + y(q)b\right\}}$$

$$= \frac{w(p)}{\sum_{q\in P} w(q)} + \frac{w(p)\max\left\{0, 1+p^T x + y(p)b\right\}}{\sum_{q\in P} w(q)\max\left\{0, 1+q^T x + y(q)b\right\}},$$

(39)

where the inequality holds by plugging $a_1 := \frac{w(p)\|x\|_2^2}{2\sum_{q\in P} w(q)}$, $r_1 := w(p)\max\left\{0, 1+p^T x + y(p)b\right\}$, $a_2 := \frac{1}{2}\|x\|_2^2$ and $r_2 = \sum_{q\in P} w(q)\max\left\{0, 1+q^T x + y(q)b\right\}$ into Claim 21, and the equality holds since $\|x\|_2 \geq \gamma$.

In addition, we observe that

$$\frac{w(p)\max\left\{0, 1+p^T x + y(p)b\right\}}{\sum_{q\in P} w(q)\max\left\{0, 1+q^T x + y(q)b\right\}} \leq \frac{w(p)\max\left\{0, 1+p^T x + y(p)b\right\}}{\sum_{q\in P_{y(p)}} w(q)\max\left\{0, 1+q^T x + y(q)b\right\}}$$

$$\leq \frac{w(p)\max\left\{0, 1+\|x\|_2 + y(p)b\right\}}{\sum_{q\in P_{y(p)}} w(q)\max\left\{0, 1-\|x\|_2 + y(q)b\right\}}$$

$$\leq \frac{w(p)\max\left\{0, 1+\frac{10y(p)}{9}b\right\}}{\sum_{q\in P_{y(p)}} w(q)\max\left\{0, 1+\frac{8y(q)}{9}b\right\}}$$

(40)

$$= \frac{w(p)\left(1+\frac{10y(p)}{9}b\right)}{\sum_{q\in P_{y(p)}} w(q)\left(1+\frac{8y(q)}{9}b\right)}$$

$$\leq \frac{w(p)}{\sum_{q\in P_{y(p)}} w(q)} + \frac{5w(p)}{4\sum_{q\in P_{y(p)}} w(q)}$$

$$= \frac{9w(p)}{4\sum_{q\in P_{y(p)}} w(q)}$$

where the first inequality holds since $P \subseteq P_{y(p)}$, the second inequality holds since $\|q\|_2 \leq 1$ for every $q \in P$, both the third inequality and the equality is by the assumption of the case, and the last inequality follows from plugging $a_1 := w(p)$, $r_1 := w(p)\frac{11y(p)}{10}b$, $a_2 := \sum_{q\in P} w(q)$, and $r_2 := \frac{8}{9}\sum_{q\in P} w(q)$ into Claim 21.

Combining (39) and (40), yields that

$$\frac{w(p)f_{\text{SVM}}(p,(x\mid b))}{\sum_{q\in P} w(q)f_{\text{SVM}}(q,(x\mid b))} \leq \frac{13w(p)}{4\sum_{q\in P_{y(p)}} w(q)}$$

(c) Otherwise, i.e., $y(p)b \leq -9\|x\|_2$, we have that for every $q \in P_{y(p)}$

$$\max\left\{0, 1+q^T x + y(q)b\right\} \leq \max\left\{0, 1+\|x\|_2 + y(q)b\right\} = 0,$$

where the first inequality holds since $\|q\|_2 \leq 1$ for every $q \in P$, and the fact that $1-8\gamma < 0$.

Thus,

$$\frac{w(p)f_{\text{SVM}}(p,(x\mid b))}{\sum_{q\in P} f_{\text{SVM}}(p,(x\mid b))} \leq \frac{w(p)f_{\text{SVM}}(p,(x\mid b))}{\sum_{q\in P_{y(p)}} w(q)f_{\text{SVM}}(q,(x\mid b))} = \frac{w(p)}{\sum_{q\in P_{y(p)}} w(q)}.$$

(41)

By combining the three cases above, we obtain that

$$s(p) \leq 2 \max \left\{ \frac{w(p)\,(1+2\gamma)}{\sum\limits_{q \in P_{y(p)}} w(q)\,(1-2\gamma)}, \frac{w(p)}{\sum\limits_{q \in P \setminus P_{y(p)}} w(q)} \right\} + \frac{13 w(p)}{4 \sum\limits_{q \in P_{y(p)}} w(q)}$$
$$+ \frac{(1+10\gamma) \sum\limits_{q \in P} w(q)}{\gamma^2 \lambda} \cdot \left( w(q)\,\|U(p)\|_2^2 + \frac{w(p)}{\sum\limits_{q \in P} w(q)} \right), \tag{42}$$

Claim (i) now holds as

$$\sum_{j=1}^{d} |U(p)e_j|^2 = \|U(p)\|_2^2,$$

where the equality follows from the definition of $e_j$ for every $j \in [d]$.

**Proof of Claim (ii).**   As for the total sensitivity, we first note that that

$$\sum_{q \in P_{y(p)}} \frac{w(q)}{\sum\limits_{q' \in P_{y(p)}} w(q')} = 1. \tag{43}$$

In addition, by Lemma 5,

$$\sum_{q \in P} w(q)\,\|U(q)\|_2^2 \leq d. \tag{44}$$

Hence,

$$\sum_{q \in P} s(q) \leq \sum_{q \in P} 2 \left( \frac{w(p)\,(1+2\gamma)}{\sum\limits_{q' \in P_{y(q)}} w(q')\,(1-2\gamma)} + \frac{w(q)}{\sum\limits_{q' \in P \setminus P_{y(q)}} w(q')} \right) + \frac{13 w(p)}{4 \sum\limits_{q \in P_{y(p)}} w(q)} \tag{45}$$
$$+ \frac{(1+10\gamma) \sum\limits_{q \in P} w(q)}{\gamma^2 C} \sum_{q \in P} \left( w(q)\,\|U(p)\|_2^2 + \frac{w(p)}{\sum\limits_{q \in P} w(q)} \right)$$
$$\leq \frac{4\,(1+2\gamma)}{1-2\gamma} + \frac{\sum\limits_{q \in P_{y(p)}} w(q)}{\sum\limits_{q' \in P \setminus P_{y(p)}} w(q')} + \frac{\sum\limits_{q' \in P \setminus P_{y(p)}} w(q')}{\sum\limits_{q \in P_{y(p)}} w(q)} + \frac{13}{2} \tag{46}$$
$$+ \frac{(1+10\gamma) \sum\limits_{q \in P} w(q)}{\gamma^2 \lambda} \cdot (d+2)$$

where (45) holds since both arguments of the max operator at (42) are non-negative and their sum exceeds the max among them, and (46) holds by combining (43) with (44)

Claim (ii) now holds by substituting $\gamma = 0.4$.                                                          $\square$

**Corollary 9** (Support Vector Machines)**.** *Let $P \subseteq \mathbb{R}^d$ be a set of $n$ points such that for every $p \in P$, $\|p\| \leq 1$. Let $y : P \to \{1, -1\}$ be a labelling function, $\lambda \geq 1$ be a regularization parameter such that for every $p \in P$, $x \in \mathbb{R}^d$, and $b \in \mathbb{R}$, $f_{\mathrm{SVM}}\left( p, \begin{bmatrix} x \\ b \end{bmatrix} \right) = \lambda \max \left\{ 0, 1 - \left( p^T x + y(p) \cdot b \right) \right\} + \frac{1}{2n} \|x\|_2^2$. Let $P_+ = \{p | p \in P, y(p) = 1\}$, $P_- = P \setminus P_+$, $\tilde{C} = \frac{|P_+|}{|P_-|}$.*

*Then, there exists an algorithm that gets the set $P$ as an input, and returns a pair $(S, v)$, such that (i) with probability at least $1 - \delta$, $(S, v)$ is an $\varepsilon$-coreset for $P$ with respect to $f_{\mathrm{SVM}}$, and (ii) the size of the coreset is $|S| \in O\left( \frac{1}{\varepsilon^2} \left( \frac{dn}{\lambda} + \frac{\tilde{C}^2 + 1}{\tilde{C}} \right) \left( d \log \left( \frac{dn}{\lambda} + \frac{\tilde{C}^2 + 1}{\tilde{C}} \right) + \log \frac{1}{\delta} \right) \right)$.*

*Proof.* First, observe that by Lemma 29 the total sensitivity of the query space $(P, w, \mathbb{R}^d, f_{\text{SVM}})$ is bounded by $O\left(\left(\frac{dC}{\lambda} + \frac{\tilde{C}^2+1}{\tilde{C}}\right)\right)$. Let $s(p)$ be the upper bound on the sensitivity of each point $p \in P$ as in Lemma 29, and let $t = \sum_{q \in P} s(q)$. Let $S$ be an i.i.d random sample of size $O\left(\frac{1}{\varepsilon^2}\left(\frac{dC}{\lambda} + \frac{\tilde{C}^2+1}{\tilde{C}}\right)\left(d\log\left(\frac{dC}{\lambda} + \frac{\tilde{C}^2+1}{\tilde{C}}\right) + \log\frac{1}{\delta}\right)\right)$, where each point $p \in P$ is sampled with probability $\frac{s(p)}{t}$, and let $v(p) = \frac{w(p)t}{s(p)|S|}$. Hence by Theorem 3, we get that with probability at least $1 - \delta$, $(S, v)$ is an $\varepsilon$-coreset for the query space $(P, w, \mathbb{R}^d, f_{\text{SVM}})$ of size $|S| \in O\left(\frac{1}{\varepsilon^2}\left(\frac{dC}{\lambda} + \frac{\tilde{C}^2+1}{\tilde{C}}\right)\left(d\log\left(\frac{dC}{\lambda} + \frac{\tilde{C}^2+1}{\tilde{C}}\right) + \log\frac{1}{\delta}\right)\right)$.

$\square$

### F.6 Proof of Corollary 10

First, we provide the following definitions.

**Definition 30** (Induced matrix norm)**.** *Let $z \in [1, \infty]$. Then the $\ell_z$ induced norm for any matrix $A \in \mathbb{R}^{d \times d}$, is defined by,*

$$\|A\|_z = \max_{\substack{x \in \mathbb{R}^d \\ \|x\|_z = 1}} \|Ax\|_z.$$

**Definition 31** (SVD factorization of a square matrix)**.** *Let $A \in \mathbb{R}^{d \times d}$ be matrix. The SVD factorization of $A$ is defined to be*

$$A = \mathcal{U}\Sigma\mathcal{V}^T,$$

*where $\mathcal{U} \in \mathbb{R}^{d \times d}$ is an orthogonal matrix, $\Sigma \in \mathbb{R}^{d \times d}$ is a diagonal matrix of non-negative entries in a descending order, i.e, for every $i, j \in [d]$ such that $i \leq j$, $\Sigma_{i,i} \geq \Sigma_{j,j}$, and finally $\mathcal{V} \in \mathbb{R}^{d \times d}$ is an orthogonal matrix.*

**Lemma 32.** *For every vector $x = (x_1, \cdots, x_d) \in \mathbb{R}^d$ there is $j \in [d]$ such that*

$$\|x\|_1 \leq \left(\frac{3d}{2} - 1\right) \cdot |x_1 + x_j|.$$

*Equality holds for $x = (1, -3, \cdots, -3)$ and every $j \in [d]$, i.e., the bound is tight.*

*Proof.* Without loss of generality, assume that $x_1 \in \{0, 1\}$. Otherwise, we divide $x$ by $x_1$.

Let $j \in \arg\max_{i \in [d]} |x_1 + x_i|$, $m \in \arg\max_{i \in [d]} |x_i|$, $a = \max_{i \in [d]} x_i$, and $b = \max_{i \in [d]} -x_i$. The proof is by case analysis of three cases: (i) $x_1 = 0$, (ii) $x_1 = 1$ and $|1 + x_j| = 1 + a$, and (iii) $x_1 = 1$ and $|1 + x_j| = b - 1$.

There are no other cases, since if $x_1 = 1$,

$$|x_1 + x_j| = |1 + x_j| = \max\{1 + x_j, -x_j - 1\} = \max\{1 + a, b - 1\}.$$

We observe that:

(i) If $x_1 = 0$,

$$\|x\|_1 = \sum_{i=1}^d |x_i| \leq d|x_m| = d|x_1 + x_m| \leq ((3d/2) - 1) \cdot |x_1 + x_m|,$$

where the last inequality is by assumption $d \geq 2$, otherwise the lemma is trivial.

(ii) If $|1 + x_j| = 1 + a$ and $x_1 = 1$, then for every $i \in [d]$,

$$|x_i| = |1 + x_i - 1| \leq |1 + x_i| + 1 \leq |1 + x_j| + 1 = 2 + a, \tag{47}$$

where the first inequality is by the triangle inequality, and the last equality holds by the assumption of the case. Hence

$$\frac{\|x\|_1}{|x_1 + x_j|} \leq \frac{1 + (d-1)(a+2)}{1 + a}. \tag{48}$$

The right hand side is decreasing with $a$ since the numerator of its derivative is

$$(d-1)(1+a) - (1 + (d-1)(a+2)) = -d < 0.$$

Its maximum is achieved at $a \geq x_1 = 1$ by the assumption $x_1 = 1$ of this case. By this and (48),

$$\frac{\|x\|_1}{|x_1 + x_j|} \leq \frac{1 + (d-1)(a+2)}{1+a} \leq \frac{1 + 3(d-1)}{2} = \frac{3d}{2} - 1.$$

(iii) $|1 + x_j| = b - 1$ and $x_1 = 1$. For every $i \in [d]$ we thus have $|x_i| \leq |1 + x_j| + 1 = b$, similarly to (47). Hence

$$\frac{\|x\|_1}{|x_1 + x_j|} \leq \frac{1 + b(d-1)}{b-1}. \tag{49}$$

The right hand side is decreasing with $b$ since the enumerator of its derivative is

$$(d-1)(b-1) - (1 + b(d-1)) = -d < 0.$$

Its maximum is achieved at $b = |1 + x_j| + 1 \geq |1 + x_1| + 1 = 3$, where the first equality is by the assumptions of Case (iii). By this and (49),

$$\frac{\|x\|_1}{|x_1 + x_j|} \leq \frac{1 + b(d-1)}{b-1} \leq \frac{1 + 3(d-1)}{2} = \frac{3d}{2} - 1.$$

$\square$

**Claim 33.** *Let $A \in \mathbb{R}^{d \times d}$ be an invertible matrix, and let $A = \mathcal{U}\Sigma\mathcal{V}$ be the SVD factorization of $A$ (see Definition 31). Then for every $i \in [2, d]$,*

$$\|A\|_2 \leq \|A(\mathcal{V}_1 + \mathcal{V}_i)\|_2,$$

*where $\mathcal{V}_j$ denotes the jth column of $\mathcal{V}$ for every $j \in [d]$.*

*Proof.* First, put $i \in [2, d]$, and note that by [46], we have that

$$\|A\|_2 = \|A\mathcal{V}_1\|_2.$$

For every $j \in [d]$, let $e_j$ denotes the vector with a 1 in the jth coordinate and 0's elsewhere. We observe that

$$\|A\mathcal{V}_1 + A\mathcal{V}_i\|_2^2 = \|A\mathcal{V}_1\|_2^2 + 2\mathcal{V}_1^T A^T A\mathcal{V}_i + \|A\mathcal{V}_i\|_2^2. \tag{50}$$

By orthogonality of $\mathcal{V}$ and $\mathcal{U}$,

$$\mathcal{V}_1^T A^T A\mathcal{V}_i = \mathcal{V}_1^T \mathcal{V}\Sigma^T \mathcal{U}^T \mathcal{U}\Sigma\mathcal{V}^T \mathcal{V}_i = \mathcal{V}_1^T \mathcal{V}\Sigma^T \Sigma\mathcal{V}^T \mathcal{V}_i = e_1^T \Sigma^T \Sigma e_i = \Sigma_{1,1}\Sigma_{i,i}e_1 e_i^T = 0, \tag{51}$$

where the first equality holds by Definition 31, the second equality is by orthogonality of $\mathcal{U}$, the third equality is by orthogonality of $\mathcal{V}$, the forth equality holds since $\Sigma$ is a diagonal matrix and the last equality holds by definition of $e_j$ for every $j \in [d]$.

Combining (50) and (51), yields that

$$\|A\mathcal{V}_1 + A\mathcal{V}_i\|_2 = \sqrt{\|A\mathcal{V}_1\|_2^2 + \|A\mathcal{V}_i\|_2^2} \geq \|A\mathcal{V}_1\|_2 = \|A\|_2.$$

$\square$

**Lemma 34.** *Let $(P, w, \mathbb{R}^d, f_{\mathrm{RES}\ell_z})$ be a query space as in Definition 11, such that for every $x \in \mathbb{R}^d$, and $p \in P$, the loss function $f_{\mathrm{RES}\ell_z}$ is defined to be*

$$f_{\mathrm{RES}\ell_z}(p, x) = \min\left\{ |p^T x|, \|x\|_z \right\}.$$

*Let $g_{\mathrm{RES}\ell_z} \in \mathcal{F}$ such that for every $x \in \mathbb{R}^d$ and $p \in P$, $g_{\mathrm{RES}\ell_z}(p, x) = |p^T x|$. Let $(U, D, V)$ be the F-SVD of $P$ with respect to $g_{\mathrm{RES}\ell_z}$. Let $\gamma = \max\left\{ 1, \frac{2\pi d^{\left|\frac{1}{2} - \frac{1}{z}\right|}}{\|DV^T\|_2} \right\}$. Then claims (i) – (ii) hold as follows:*

(i) *For every $p \in P$, its sensitivity with respect to the query space $(P, w, \mathbb{R}^d, f_{\text{RES}\ell_z})$ is bounded by*

$$s(p) = w(p) \min \left\{ \|U(p)\|_2 , d^{|\frac{1}{2} - \frac{1}{z}|} \left\| (DV^T)^{-1} \right\|_2 \right\},$$

(ii) *and the total sensitivity is bounded by*

$$\sum_{p \in P} s(p) \le 4\gamma d^{2 + |\frac{1}{2} - \frac{1}{z}|}.$$

*Proof.* First, we observe that the level set $\mathcal{X}_{g_{\text{RES}\ell_z}}$ (see Definition 1) is contained in the level set $L = \left\{ x \,\middle|\, x \in \mathbb{R}^d, \sum_{p \in P} w(p) f_{\text{RES}\ell_z}(p, x) \le 1 \right\}$. By Theorem III of [36], the Löwner ellipsoid which contains the level set $\mathcal{X}_{g_{\text{RES}\ell_z}}$ will also contain the level set $L$, when setting the dilation factor, i.e., $\alpha$ to $\gamma\sqrt{d}$. In other words,

$$\frac{1}{\sqrt{d}} E \subseteq \mathcal{X}_{g_{\text{RES}\ell_z}} \subseteq L \subseteq \sqrt{d}\gamma E,$$

where $E$ denotes the Löwner ellipsoid of the level set $\mathcal{X}_{g_{\text{RES}\ell_z}}$. Since $L$ is contained in the ellipsoid $\sqrt{d}\gamma E$, and contains the ellipsoid $\frac{1}{\sqrt{d}} E$, using similar arguments to those established at the proof of Lemma 16, we obtain that there exists a diagonal matrix $D \in \mathbb{R}^{d \times d}$ and an orthogonal matrix $V \in \mathbb{R}^{d \times d}$ such that for every $x \in \mathbb{R}^d$,

$$\left\| D'V^T x \right\|_2 \le \sum_{q \in P} w(q) f_{\text{RES}\ell_z}(q, x) \le \gamma\sqrt{d} \left\| D'V^T x \right\|_2, \tag{52}$$

where $D' := \frac{1}{2\gamma} D$.

With this, we proceed to bound the sensitivity of each point $p \in P$.

**Proof of Claim (i).** Put $p \in P$, and let $U(q) := (VD')^{-1} q$ for every $q \in P$. Observe that,

$$\sup_{\substack{x \in \mathbb{R}^d \\ f_{\text{RES}\ell_z}(p,x) > 0}} \frac{w(p) f_{\text{RES}\ell_z}(p, x)}{\sum_{q \in P} w(q) f_{\text{RES}\ell_z}(q, x)} \le \sup_{\substack{x \in \mathbb{R}^d, \\ f_{\text{RES}\ell_z}(p,x) > 0}} \frac{w(p) f_{\text{RES}\ell_z}(p, x)}{\|D'V^T x\|_2}$$

$$= \sup_{\substack{x \in \mathbb{R}^d, \\ f_{\text{RES}\ell_z}(p,x) > 0}} w(p) \min \left\{ \frac{|U(p)^T D'V^T x|}{\|D'V^T x\|_2}, \frac{\|x\|_z}{\|D'V^T x\|_2} \right\}$$

$$\le w(p) \min \left\{ \|U(p)\|_2 , d^{|\frac{1}{2} - \frac{1}{z}|} \left\| (D'V^T)^{-1} \right\|_2 \right\} \tag{53}$$

where the first inequality is by (52), the equality is by definition of $f_{\text{RES}\ell_z}$, and the last inequality follows from combining Lemma 18 with the fact that $\frac{D'V'^T x}{\|D'V'^T x\|_2}$ is a unit vector and $(D'V^T)^{-1} D'V^T = I_d$.

**Proof of Claim (ii).** In order to bound the total sensitivity, we first let $\beta_z = d^{|\frac{1}{2} - \frac{1}{z}|}$, $M \in \mathbb{R}^{d \times d}$ be an orthogonal matrix that corresponds to the matrix $\mathcal{V}$ of the *SVD* factorization of $(D'V^T)^{-1}$ (See Definition 31), and let $M_i$ denote the $i$th column of $M$ for every $i \in [d]$. Thus,

$$\min \left\{ \|U(p)\|_2 , \beta_z \left\| (D'V^T)^{-1} \right\|_2 \right\} = \min \left\{ \|MU(p)\|_2 , \beta_z \left\| (D'V^T)^{-1} M_{1*} \right\|_2 \right\}$$

$$\le \min \left\{ \|MU(p)\|_1 , \beta_z \left\| (D'V^T)^{-1} M_{1*} \right\|_2 \right\}$$

$$\le \min \left\{ 2d \left| U(p)^T (M_1 + M_j) \right| , \beta_z \left\| (D'V^T)^{-1} (M_1 + M_j) \right\|_2 \right\}$$

$$\le 2d \min \left\{ \left| U(p)^T (M_1 + M_j) \right| , \left\| (D'V^T)^{-1} (M_1 + M_j) \right\|_2 \right\}, \tag{54}$$

where the equality holds by definition of $M$, the first inequality holds by Lemma 18, the second inequality holds by Lemma 32 and by Claim 33, and the last inequality follows from the fact that $\beta_z \leq 2d$.

By combining (53) and (54), we have that

$$s(p) \leq 2d^{1+\left|\frac{1}{2}-\frac{1}{z}\right|} w(p) \sum_{j=1}^{d} \min\left\{ \left| U(p)^T (M_{*1} + M_{*j}) \right|, \left\| \left( D'V^T \right)^{-1} (M_{*1} + M_{*j}) \right\|_z \right\}, \quad (55)$$

where the inequality follows from invoking Lemma 18.

Summing (55) over every $p \in P$, we obtain that

$$\sum_{p \in P} s(p) \leq 2\gamma d^{1+\left|\frac{1}{2}-\frac{1}{z}\right|} \sum_{j=1}^{d} \|M_1 + M_j\|_2 \leq 4\gamma d^{2+\left|\frac{1}{2}-\frac{1}{z}\right|},$$

where the first inequality is by (52) and the second inequality holds since $\|x+y\|_2 \leq 2$ for any pair of unit vectors $x, y \in \mathbb{R}^d$. $\qquad\square$

**Corollary 10** (Outlier resistant functions). *Let $P \subseteq \mathbb{R}^d$ be a set of $n$ points, and let $f_{\mathrm{RES}\ell_z} : P \times \mathbb{R}^d \to [0, \infty)$ be loss function such that for every $x \in \mathbb{R}^d$, and $p \in P$, $f_{\mathrm{RES}\ell_z}(p, x) = \min\left\{ \left| p^T x \right|, \|x\|_z \right\}$.*

*Then, there exists an algorithm that gets the set $P$ as an in input, and returns a pair $(S, v)$, such that (i) with probability at least $1 - \delta$, $(S, v)$ is an $\varepsilon$-coreset for $P$ with respect to $f_{\mathrm{RES}\ell_z}$, and (ii) the size of the coreset is $O\left( \frac{\gamma d^{2+\left|\frac{1}{2}-\frac{1}{z}\right|}}{\varepsilon^2} \left( d \log\left( \gamma d^{2+\left|\frac{1}{2}-\frac{1}{z}\right|} \right) + \log\left(\frac{1}{\delta}\right) \right) \right)$, where $\gamma$ is defined in the proof.*

*Proof.* First, observe that by Lemma 34 the total sensitivity of the query space $(P, w, \mathbb{R}^d, f_{\mathrm{RES}\ell_z})$ is bounded by $O\left( \gamma d^{2+\left|\frac{1}{2}-\frac{1}{z}\right|} \right)$. Let $s(p)$ be the upper bound on the sensitivity of each point $p \in P$ as in Lemma 34, and let $t = \sum_{q \in P} s(q)$. Let $S$ be an i.i.d random sample of size $O\left( \frac{\gamma d^{2+\left|\frac{1}{2}-\frac{1}{z}\right|}}{\varepsilon^2} \left( d \log\left( \gamma d^{2+\left|\frac{1}{2}-\frac{1}{z}\right|} \right) + \log\frac{1}{\delta} \right) \right)$, where each point is sampled with probability $\frac{s(p)}{t}$, and let $v(P) = \frac{w(p)t}{s(p)|S|}$. Hence by Theorem 3, we get that with probability at least $1 - \delta$, $(S, v)$ is an $\varepsilon$-coreset for the query space $(P, w, \mathbb{R}^d, f_{\mathrm{RES}\ell_z})$.

$\qquad\square$

# G  "Easy" examples covered by our framework

## G.1  $\ell_z$-Regression for $z \in [1, \infty)$

**Lemma 35.** *Let $z \in [1, \infty)$, $(P, w, \mathbb{R}^d, f_{\ell_z})$ be a query space, such that fo every $x \in \mathbb{R}^d$ and $p \in P$ the loss function $f_{\ell_z} : P \times \mathbb{R}^d \to [0, \infty)$ is defined to be $f_{\ell_z}(p, x) = \left| p^T x \right|^z$. Let $(U, D, V)$ be the f-SVD of $(P, w)$ with respect to $f_{\ell_z}$ (see Definition 4). Then, claims (i) – (ii) hold as follows:*

*(i) for every $p \in P$, the sensitivity of $p$ with respect to the query space $(P, w, \mathbb{R}^d, f_{\ell_z})$ is bounded by*

$$s(p) \leq \begin{cases} w(p) \|U(p)\|_z^z & z \in [1, 2] \\ \sqrt{d^z} w(p) \|U(p)\|_z^z & otherwise \end{cases},$$

*(ii) and the total sensitivity is bounded by*

$$\sum_{p \in P} s(p) \leq \begin{cases} d^{\frac{z}{2}+1} & z \in [1, 2) \\ d & z = 2 \\ d^{z+1} & otherwise \end{cases}.$$

*Proof.* Note the following:

(a) For every $q \in P$, $x \in \mathbb{R}^d$ and $b \geq 0$ we have $\left|q^T b x\right| = b \left|q^T x\right|$.

(b) Since $\left|q^T x\right|$ is convex function, it also holds that $\sum_{q \in P} w(q) \left|q^T x\right|^z$ is convex due to the fact that sum of convex functions is also convex,

(c) The level set $\left\{ x \middle| x \in \mathbb{R}^d, \sum_{q \in P} w(q) \left|q^T x\right|^z \leq 1 \right\}$ is convex and is centrally symmetric.

(d) For any unit vector $x \in \mathbb{R}^d$ and $q \in P$,

$$\left|U(q)^T x\right|^z \leq \|U(q)\|_2^z \|x\|_2 = \|U(q)\|_2^z \leq \begin{cases} \|U(q)\|_z^z & z \in [1, 2] \\ d^{\frac{1}{2} - \frac{1}{z}} \|U(q)\|_z^z & z > 2 \end{cases},$$

where the first inequality holds by Cauchy Schwartz's inequality, the equality is by the assumption that $x$ is a unit vector, and the last inequality holds by plugging $a := 2$ and $b := z$ for $z > 2$ and $a := z$ and $b := 2$ for $z \in [1, 2]$ into Lemma 18.

Hence, plugging

- $f(p, x) := f_{\ell_z}(p, x)$, $g(p, x) := f_{\ell_z}(p, x)$, $h(p, x) := 0$ for every $p \in P$, and $x \in \mathbb{R}^d$,

- $c_i := 1$ for every $i \in [5]$,

- $z := z$,

- $\alpha := \sqrt{d}$ for $z \neq 2$ and $\alpha := 1$ for $z = 2$,

- $v_i := e_i$ where $e_i$ denotes a vector which at its $i$th entry there is 1, and 0's elsewhere,

- and $c := \begin{cases} 1 & z \in [1, 2] \\ d^{\frac{1}{2} - \frac{1}{z}} & z > 2 \end{cases}$,

into Lemma 5, yields that

$$s(p) = w(p) \sum_{i \in [d]} \left|U(p)^T D V^T e_i\right|^z \cdot \begin{cases} 1 & z \in [1, 2] \\ d^{\frac{1}{2} - \frac{1}{z}} & z > 2 \end{cases}$$

This satisfies (i) as

$$\sum_{i \in [d]} \left|U(q)^T e_i\right|^z = \|U(q)\|_z^z,$$

holds for every $q \in P$ by definition of norms.

As for the sum of sensitivities, Claim (ii) follows from Lemma 5. $\qquad \square$

**Corollary 36.** *Let $(P, w, \mathbb{R}^d, f_{\ell_z})$ be a query space, such that for every $x \in \mathbb{R}^d$, and $p \in P$, the loss function $f_{\text{NC}\ell_z}$ is defined to be*

$$f_{\ell_z}(p, x) = \left|p^T x\right|^z.$$

*Let $\varepsilon, \delta \in (0, 1)$, and let $(S, v)$ be the output of a call to CORESET $(P, w, f_{\ell_z}, \varepsilon, \delta)$. Then, with probability at least $1 - \delta$, $(S, v)$ is an $\varepsilon$-coreset for the query space $(P, w, \mathbb{R}^d, f_{\ell_z})$, and the size of the coreset is*

$$|S| \in \begin{cases} O\left(\frac{d^{\frac{z}{2} + 1}}{\varepsilon^2} \left(d \log\left(d^{\frac{z}{2} + 1}\right) + \log\left(\frac{1}{\delta}\right)\right)\right) & z \in [1, 2) \\ O\left(\frac{d}{\varepsilon^2} \left(d \log(d) + \log\left(\frac{1}{\delta}\right)\right)\right) & z = 2 \\ O\left(\frac{d^{z+1}}{\varepsilon^2} \left(d \log\left(d^{z+1}\right) + \log\left(\frac{1}{\delta}\right)\right)\right) & z \in (2, \infty) \end{cases}.$$

*Proof.* First, observe that by Lemma 35, the total sensitivity is bounded by $t := \begin{cases} d^{\frac{z}{2}+1} & z \in [1,2) \\ d & z = 2 \\ d^{z+1} & \text{otherwise} \end{cases}$.

Plugging $s(p)$ for every $p \in P$ from Lemma 35, $t := t$, $\varepsilon := \varepsilon$ and $\delta := \delta$ into Theorem 6, yields that with probability at least $1 - \delta$, $(S, v)$ is an $\varepsilon$-coreset of size $O\left(\frac{t}{\varepsilon^2}\left(d\log(t) + \log\left(\frac{1}{\delta}\right)\right)\right)$. $\qquad\square$

### G.2 Least squared errors

**Lemma 37.** *Let $(P, w, \mathbb{R}^d, f_{\text{LSE}})$ be a query space, such that for every $x \in \mathbb{R}^d$ and $p \in P$, the loss function $f_{\text{LSE}}$ is defined to be $f_{\text{LSE}}(p, x) = \|p - x\|_2^2$. Let $P' = \left\{ p' = \begin{bmatrix} \|p\|_2^2 \\ 2p \\ 1 \end{bmatrix} \middle| p \in P \right\}$ and let*

$g_{\text{LSE}} : P' \times \mathbb{R}^{d+2} \to [0, \infty)$ *such that for every $y \in \mathbb{R}^{d+2}$ and $p \in P'$, $g_{\text{LSE}}(p, y) = |p^T y|$. Let $(U, D, V)$ be the $f$-SVD of $P'$ with respect to $g_{\text{LSE}}$. Then, claims $(i) - (ii)$ hold as follows:*

(i) *for every $p \in P$, the sensitivity of $p$ with respect to the query space $(P, w, \mathbb{R}^d, f_{\text{LSE}})$ is bounded by $s(p) \leq w(p) \|U(p')\|_1$,*

(ii) *and the total sensitivity is bounded by $\sum_{p \in P} s(p) \in O\left(d^{1.5}\right)$.*

*Proof.* Put $p \in P$, and observe that for every $y \in \mathbb{R}^d$, $\|p - y\|_2^2 = \|p\|_2^2 - 2p^T y + \|y\|_2$, which enables us to rewrite the problem by reformulating the query space and the input space ($\mathbb{R}^d$ and $P$ respectively). Let $X' = \left\{ \begin{bmatrix} 1 \\ -x \\ \|x\|_2^2 \end{bmatrix} \middle| x \in \mathbb{R}^d \right\}$. Then, we obtain that for every $x \in X'$

$$\frac{w(p) f_{\text{LSE}}(p, x)}{\sum_{q \in P} w(q) f_{\text{LSE}}(q, x)} = \frac{w(p) \left|p'^T x\right|}{\sum_{q \in P} w(q) \left|q'^T y\right|} \leq \sup_{y \in \mathbb{R}^{d+2}} \frac{w(p) \left|p'^T y\right|}{\sum_{q \in P} w(q) \left|q'^T y\right|},$$

where the second inequality is by rewriting the cost function and setting $y \in X'$ and the last inequality follows from $\sup$ operator.

Finally, the upper bound on the sensitivity of each point $p \in P$ and an upper bound on the total sensitivity follows from plugging $P'$, $\mathbb{R}^{d+2}$ as the query space, and $z := 1$ into Corollary 35. $\qquad\square$

**Corollary 38.** *Let $(P, w, \mathbb{R}^d, f_{\text{LSE}})$ be a query space, such that for every $x \in \mathbb{R}^d$, and $p \in P$, the loss function $f_{\text{LSE}}$ is defined to be*

$$f_{\text{LSE}}(p, x) = \|p - x\|_2^2.$$

*Let $P' = \left\{ \begin{bmatrix} p' = \|p\|_2^2 \\ 2p \\ 1 \end{bmatrix} \middle| p \in P \right\}$ and let $g_{\text{LSE}} : P' \times \mathbb{R}^{d+2} \to [0, \infty)$ such that for every $x \in \mathbb{R}^{d+2}$ and $p \in P'$, $g_{\text{LSE}}(p, x) = |p^T x|$. For every $p \in P$ and $p' = (\|p\|_2^2 \mid 2p \mid 1)$ we define $w'(p') = w(p)$. Let $\varepsilon, \delta \in (0, 1)$, and let $(S', v')$ be a coreset for the query space $\left(P', w', \mathbb{R}^{d+2}, g_{\text{LSE}}\right)$ by Corollary 36. Let $S = \left\{ p \mid (\|p\|_2^2 \mid 2p \mid 1) \in S' \right\}$, and for every $p \in S$, and $p' = (\|p\|_2^2 \mid 2p \mid 1) \in S'$ let $v(p) = v'(p')$. Then, with probability at least $1 - \delta$, $(S, v)$ is an $\varepsilon$-coreset for the query space $(P, w, \mathbb{R}^d, f_{\text{LSE}})$, and the size of the coreset is $|S| \in O\left(\frac{(d+2)^{2.5}}{\varepsilon^2}\left(d\log\left((d+2)^{2.5}\right) + \log\left(\frac{1}{\delta}\right)\right)\right)$.*

*Proof.* First, observe that by Lemma 37, the total sensitivity is bounded by $t := (d+2)^{1.5}$ of $\left(P', w', \mathbb{R}^{d+2}, g_{\text{LSE}}\right)$. Plugging $P := P'$, $t := t$, $\varepsilon := \varepsilon$ and $\delta := \delta$ into Corollary 36, yields that $S', v'$ is an $\varepsilon$-coreset of size $O\left(\frac{(d+2)^{2.5}}{\varepsilon^2}\left(d\log\left((d+2)^{2.5}\right) + \log\left(\frac{1}{\delta}\right)\right)\right)$ for the query space $\left(P', w', \mathbb{R}^{d+2}, g_{\text{LSE}}\right)$.

By construction of $P'$, it holds that for every $p' \in S'$ and $x' = \begin{bmatrix} 1 \\ -x \\ \|x\|_2^2 \end{bmatrix}$ where $x \in \mathbb{R}^d$,

$$v\left(p'\right) g_{\text{LSE}}(p', x') = v\left(p'\right) |p'x'| = v\left(p'\right) \|p - x\|_2^2 = v(p) \|p - x\|_2^2 = v(p) f_{\text{LSE}}(p, x).$$

Thus we obtain that for every $x \in \mathbb{R}^d$

$$\left| \sum_{p \in P} w(p) \|p - x\|_2^2 - \sum_{p \in S} w(p) \|p - x\|_2^2 \right| \leq \varepsilon \sum_{p \in P} w(p) \|p - x\|_2^2,$$

hold with probability at least $1 - \delta$, i.e., $(S, v)$ is an $\varepsilon$-coreset for the query space $\left(P, w, \mathbb{R}^d, f_{\text{LSE}}\right)$. $\quad\square$

## H    Experimental setup

**Preprocessing step.**    We applied a standardization step, i.e., each input point has zero mean and unit variance. In addition, specifically for the problem of *SVM* and *Logistic regression*, the points were normalized such that the maximal norm of a point in the dataset will be 1.

**Faster algorithms for computing the $f$-SVD**    Problems which can be reduced to the $\ell_2$-regression problem, are easier to deal with, since the $f$-SVD can be computed using the SVD factorization which is can be computed in $O\left(n^2 d\right)$, e.g., we showed that both logistic regression and SVM can be reduced to $\ell_2$-regression as discussed in Lemma 25 and Lemma 28.

As for our aforementioned problems, we shown a reduction to $\ell_1$ regression, which using [15], we can compute the $f$-SVD in roughly $O\left(nd + poly(d)\right)$ time (worst case scenario).

Note that [15] can accelerate the computation time of the $f$-SVD if the problem can be reduced to $\ell_z$ regression for any $z \geq 1$, due to the fact that it computes an approximated Löwner ellipsoid using randomized algorithm. For other problems, the time needed for computing the $f$-SVD is mentioned at Theorem 6.