[Reviews · NeurIPS 2020]

Review 1

Summary and Contributions: This paper deals with constructing coresets in the sensitivity framework. Many such contributions focus on one specific problem, but in this paper we deal with a fairly general class of "near-convex" functions which unifies several specific problems like regularized logistic regression, regularized SVM, L_z regression, and a class of outlier resistent functions (minimum over L_z and L_1). - the definition of "near-convex" functions f(p,x) enforces that up to constants, f can be expressed powers of a convex function g and another function h, so that f is approximately g^z+h^z. Also g(p,x) needs to be linear in its argument x which is usually the (regression) parameter. h^z needs to have sensitivity c/n, basically this means h(p,x) it is uniform in p, i.e. independent of the input points p (e.g. a regularization term |x|^2 ). The most important property is that the level set of g^max(z,1) forms a bounded, centered convex set. - the last property ensures the existence of a lowler-john ellipsoid that yields a basis for the input, which is "well-conditioned" for approximating the function g. From such a basis one can bound the sensitivities of the input points in terms of their squared euclidean norm similar to the leverage scores in L_2 regression obtained via SVD. Such arguments were known before for L_p regression, and are generalized here for the broader class of "near convex" functions f. - as for applications this framework generalizes existing work on L_p regression, and regularized versions of SVM and logistic regression. In addition, the authors claim the first coresets for L_p regression for p \in (0,1) and for a class of outlier resistent functions (minimum over L_z and L_1) which they call M-estimators. - There are also some empirical results and an open implementation. In general I like this paper and I tend to recommend acceptance. However I am missing a critical discussion of the necessities of the assumptions in Definition 1 and the limitations that they impose. Also, I have some comments on the related work, see below. ########################## Dear authors, thanks for your clarifying response which encourages my initial rating. Please make sure to incorporate the promised edits on the related work and improvements on the experimental part into the final version. Some final remarks: - I think the added explanation of Definition 1 is appropriate. The only thing I am missing wrt (iv) is that the level sets of g are not only assumed to be convex, but their *boundedness* plays a crucial role that needs to be mentioned (cf. my comment on the unboundedness wrt the plain logistic and hinge loss functions) - regarding SVM you might consider choosing a more restrictive value of lambda maybe dependent on the input (similar to [53]) to get around assumptions on the balance of classes. Or you might consider an assumption on the balance of miss-classifications vs. correct-classifications (similar to [44]) to get around assuming the regularization. Best

Strengths: - A unified treatment of a quite general family of functions in the sensitivity framework. This is challenging non-trivial work. - generalization of well conditioned bases and SVD to this class - results for l_p, regularized logstic regression and SVM are comparable with the state of the art (not weakened by the generality), new results for l_p, p\in(0,1) and min{|x|_z,|x|_1}

Weaknesses: - I think the definition of "near convex" is quite restrictive towards the necessities of the sensitivity framework. However, it is presented as a very general and natural class of functions. The limitations are not clearly discussed in my opinion. - related work: see below - the SVM result depends on regularization and additionally on the structure of the data which seems quite restrictive to rely on both relaxations.

Correctness: I believe the results are correct. I looked into most of the technical proofs in the supplement, though not all due to time constraints. One point made me suspicious on which I found no discussion. Please comment on that point: To my knowledge L_z, z \in (0,1) has non-convex level sets and even if an enclosing and enclosed ellipsoid could be found, the scaling gap might be larger than \Theta(sqrt(d)); I guess even arbitrary larger as p -> 0. Can you explain your solution to this intuitively, especially why it is sufficient to use the convex L_1 level set instead and "outsource" the power of z .

Clarity: Yes

Relation to Prior Work: Here I found some issues where correction is needed: 1. The paper suggests that including an intercept term and regularization are more general than previous works that considered only the linear case or were missing regularization. - In fact the linear case is more general, since including the intercept is only the special case where one coordinate is fixed to 1. - Also the unregularized versions of either logistic regression or SVM are more general wrt coresets because any coreset for the plain problems is also a coreset when adding any non-negative regularization. The reverse is not necessarily true. - Finally note that coresets for the unregularized versions have linear lower bounds [44,53], while in the regularized versions studied in this paper (with lambda=sqrt(n)) allow coresets of size O(sqrt(n)) by simple uniform subsampling [17], which indicates that the regularized version is a significantly simpler problem (a strong relaxation). 2. Previous works are "criticized" for using squared loss functions whereas in this paper the simple (more robust and practical) hinge loss is used. - It should be noted here that introducing the squared Euclidean regularizer turns the linear function into a square function again. Why should this be any better or any more natural. In my opinion this does a very similar simplification of the more complicated linear problem. - note that the applicability of the "near convex" framework relies on the regularization, because plain hinge loss (and logistic loss) have convex but unbounded level sets. Thus there are no ellipsoids as required here. 3. regarding outlier resistant functions (M-estimators), the authors claim the first coresets. Maybe this is true for the explicit function studied here but it is not true that there are no previous coresets for such functions! Just to name some: l_1 regression is outlier resistant (and even fits the definition in the paper for z=1): - http://citeseerx.ist.psu.edu/viewdoc/summary?doi=10.1.1.145.6449 - https://dl.acm.org/doi/10.1145/1993636.1993736 coresets for other M-estimators: - http://www.cs.cmu.edu/afs/cs/user/dwoodruf/www/cw15.pdf (huber loss) - http://www.cs.cmu.edu/afs/cs/user/dwoodruf/www/CW15b.pdf (more general M-estimators) - https://arxiv.org/abs/1905.05376 (Tukey loss) very similar to the "minimum definition" studied here, but more complicated.

Reproducibility: Yes

Additional Feedback: - 37: "industry and academy" none of the referenced papers seem to relate to the industry. They are all academic papers. - 44: near linear time even for NP hard prolems: It should be noted that those are exponential in 1/eps or alike... 86: in the denominator h(p,x) -> h(q,x) 85: linearity of g and 86: <=2/n seems very restrictive! 135: we also generalize -> we also unify (would fit better here) 159: Lipshcitz -> Lipschitz Figure 1 is squeezed, below "dilated" -> "contracted" Thm 3: the VC dimension should depend on the weighting, triplet -> quadruple (more occurrences below) - the existence of the set of vectors v_j should be discussed somewhere around lemma 5 and algorithm 1 - "we, we compute" - "a a set" - "we no show" - The experiments do not seem to be too much better than uniform sampling. And the construction takes more time, so it would be good to plot time vs accuracy to see whether it actually outperforms uniform sampling.


Review 2

Summary and Contributions: The paper defines the notion of near-convex functions and designs an algorithm to find a coreset for such functions. The examples of near-convex functions considered in the paper are logistic regression, L_z regression, and SVM. A function f(p,x) is near-convex if there exist a convex function g, a function h, and a scalar z such that 1) f(p,x) = \theta(g(p,x)^z +h(p,x)^z), 2) g is linear in x, 3) for every p and x, h(p,x)^z/(\sum_{q\in P} h(q,x)^z) <= 2/n, 4) X_g:={x | \sum_{p\in P} g(p,x)^{max{1,z}}<=1} is centrally symmetric and it contains a ball of radius r and is contained in a ball of radius R. The main tool used in the paper is called f-SVD factorization which refers to finding a diagonal matrix D and an orthogonal matrix V such that ||DV^T x||_2 approximates the total loss \sum_{p \in P} f(p,x) within a factor of \sqrt{d}. Then the algorithm calculates the sensitivity of each point p according to the f-SVD factorization and then samples points from P (to construct the coreset) with probabilities proportional to sensitivities.

Strengths: The notion of near-convex functions and f-SVD factorization are interesting. This approach tries to generalize and unify the process of finding a coreset for different functions and settings. Also, it somewhat decreases the dependence of the size of the coreset on the structure of the input. For example, for logistic regression, the size of the coreset is completely independent of the structure of the input, and for SVM, the size only depends on the ratio of the number of points in each class. In summary, the theoretical and conceptual contribution of the paper is noteworthy.

Weaknesses: I find the experimental results very insignificant and inconclusive. The proposed method is only compared to picking the coreset uniformly at random. It would be more interesting to compare the method to prior methods in the literature. Also, some parts of the paper are a bit vague. For example, do we assume that g and h are given? If not, how do we find them? If yes, what would be the time complexity of finding them. Moreover, some more explanation on finding matrices D and V would be helpful. It seems that it is done using Lowner ellipsoid and it is discussed in the supplementary information, but I think it would be useful to have a discussion about this in the main text of the paper.

Correctness: I did not check the proofs in the supplementary information by the claims and method seem correct.

Clarity: The paper is generally well written but there are some vague parts that I mentioned above. Moreover it might be helpful to discuss an example after Definition 1. For example what are g, h, and the constants for SVM.

Relation to Prior Work: Yes, I think the paper discuuses the relation to prior work very good. However it lacks a clear comparison with prior methods in terms of experimental results and reporting the time complexity and coreset size for prior methods in the literature.

Reproducibility: Yes

Additional Feedback: 1) typo. (line 86) the denominator of the left side of the inequality should be \sum_{q \in P} h(q, x)^z 2) In Figure 1 (iv) and (v), X_g is not centrally symmetric, but in the definition it is supposed to be. 3) The note in lines 195-199 refers to properties in Definition 4 by numbers, but this properties are not numbered in the definition. ======= Post Rebuttal Edit ======= After considering the submitted rebuttal and other reviewers' opinion, I have increased my score by one because I am convinced this paper is more theoretical. However, I still think that the authors should add more meaningful experiments to the final version.


Review 3

Summary and Contributions: Coreset is a basic primitive to solve many optimization problems. However, for discrete optimization problems, we have different coreset techniques depending on the nature of problems. This paper proposes a generic coreset technique for the class of convex funtions using the nice bridge that they make between the Löwner ellipsoid and the sensitivity sampling (well, I prefer to call it importance sampling which is known in machine learning for a long time than the sensitivity sampling). Main idea: We need to do importance sampling to bound the variance for which we need to find an upper-bound for the importance of each point. If the loss function that we would like to approximate (query function) is convex, we can sandwitch it for each point using two ellipsoids whose volumes are within sqrt(d) of each other based on the Löwner ellipsoid. This gives us upper-bound and lower-bounds for the importance sampling of each point and respectively, for the totall loss of a point set P to within sqrt(d) where d is the dimension of the ambient space. The same technique has been used by Sariel Har-Peled when he for the first time introduced the relative approximaiton for SVD. Here is the link: https://arxiv.org/pdf/1410.8802.pdf I will be happy to see this paper accepted.

Strengths: The general coreset technique for convex functions is nice.

Weaknesses: This paper is an interesting paper. The writeup of the paper needs to be improved. 1- Definition 1 is very long and confusing. It would be good if they can simplify it. 2- Figure 2 is nice. It would be good if they bring it to the beginning of the paper and explain the intuition better. 3- Theorem 3, Lemma 5, and Theorem 6 are very long. It would be good if they can simplify them.

Correctness: The paper is correct.

Clarity: The writeup can be improved, but it's not in a bad shape at the moment. However they can sell better this nice result.

Relation to Prior Work: Yes.

Reproducibility: Yes

Additional Feedback:


Review 4

Summary and Contributions: This paper presents a general framework for creating coresets for what they call: "near-convex functions." These are basically functions which can be relatively approximated by the sum of two simpler functions g & h; the first one g has some linearity properties and is centrally symmetric and bounded, the other h should be is fairly uniform. Although not explicitly described, these seem to correspond with several standard cost functions that arise in machine learning for logistic regression, SVM, and l_z-regression, where g is the standard cost part, and h is an offset or regularizer term. [l_z-regression seems to uses z instead of p so as not to overload p] The standard loss function components typically corresponds with the g term, but it seems the most general versions need a separate component to deal with the h term. And so the algorithm needs to have some extra elements to handle the full generality captured in the h term. The other key idea is that the multi-dimensionsional objective function, given its centrally-symmetric and other properties, can be approximated with an SVD sketch, and archives relative error through Lowener ellipsoid properties. The main concrete results of this paper is specific problems (logistic regression, SVM, l_z-regression) are hard to compare, because they involve subtle modeling changes. * for l_z-regression, this is first technique to handle coresets for z in (0,1). * for SVM, it allows hinge loss, not just squared hinge loss as in previous work * for logistic regression it claims to be the first to allow the offset term in the formulation. The main theorems are all written very elaborately, and when written precisely, this difference is not really explained (on in early discussion on related work) so its hard to fully grasp all of these differences. But these seem like important improvements. There are experiments, but they only compare to uniform sampling, not previous methods for similar problem formulations. These methods do show clear improvement over uniform sampling.

Strengths: This paper increases the scope of coresets for several real problems in data analysis. It presents a general technique towards these and other problems.

Weaknesses: The writing is very complex, and definitions are not well-separated from the theorems. This perhaps leads to a lack of precise discussion on subtle changes in definitions and how they improve upon prior work. The experiments only compare to random sampling, not prior work.

Correctness: They appear correct.

Clarity: It is pretty well written, but could be improved.

Relation to Prior Work: Mostly. But I think it could be more clear and more precise.

Reproducibility: Yes

Additional Feedback: The Figure captions are in a font that is too small!!! In particular Figure 1 has a nice overview of what is going on, and part of this is explained in an 8 line caption, which is in very small font -- please make this larger.

[Author Response · NeurIPS 2020]

**Addressing all reviewers:** We thank the reviewers for their careful review of our paper. Following the reviewers
comments, we: (1) have made the write-up of the paper better, (2) fixed all typos and comments from the additional
feedback sections of the review, (3) added more graphs that compare our result to prior work, and additional graphs
that compare time vs accuracy against uniform sampling to emphasize our improvements as requested, (4) added an
explanation to our definitions, theorems and figures to make things more clear, specifically speaking Definition 1 and
Figure 1, (5) discussed the differences between our result and previous works.

**Reviewer 1:** We thank the reviewer for his constructive review and detailed comments.

**Q**: Many such contributions focus on one specific problem, but in this paper we deal with a fairly general class. This is
challenging non-trivial work. **A**: This is the main contribution of the paper. We thank the reviewer for pointing this out.

**Q**: I am missing a critical discussion of the necessities of the assumptions in Definition 1 and the limitations that they
impose. **A**: Added to the introduction. In short: Assumptions (i)-(iii) in Definition 1 are used to reduce the problem to
dealing with an "easier" pair of functions where the first is a convex "bi-log-log-Lipschitz" function "g" and the second
function "h" being independent of the input points. Finally, assumption (iv) ensures that the ellipsoid which encloses
the level set of g (the convex function) exists and to be centered at the origin to avoid dealing with the center throughout
the technical proofs. Combining the properties associated with the level set of g (the convex function) and Assumptions
(i)-(iv), allow us to bound the loss function from above and below by the mahalanobis distance with respect to the
enclosing ellipsoid. This is due to the fact that the level set encloses a contracted version of the ellipsoid that encloses
the level set of g. As future work, we aim to relax the aforementioned assumptions and use the center of the enclosing
ellipsoid, since we think that this step will widen the applicability of our framework. What do you think?

**Q**: The SVM result depends on regularization and additionally on the structure of the data which seems quite restrictive
to rely on both relaxations. **A**: The SVM problem formulation presented quite an obstacle when trying to show that the
loss function is near convex. As presented in the proof of Lemma 29 in the supplementary material, we had to split the
query space into two parts, where the first lead to having a bound on the sensitivity as a function of the structure of the
data, while the latter was handled using our $f$-SVD. We think that relaxing our assumptions in Definition 1, will yield a
tighter upper bound on the sensitivity of each input point in $P$ and in turn on the total sensitivity.

**Q**: Can you explain your solution to this intuitively, especially why it is sufficient to use the convex $\ell_1$ level set instead
and "outsource" the power of $z$. **A**: First let $\tilde{P} \in \mathbb{R}^{|P| \times d}$ be a matrix such that every point in P is a row in $\tilde{P}$. Now
observe that for any $z \in (0,1)$, $\sum_{p \in P} |p^T x|^z = \left\| \tilde{P}x \right\|_z^z \geq \left\| \tilde{P}x \right\|_1^z$, where the equality holds by definition of $\tilde{P}$ and the
inequality follows from properties of norms. Using this we can bound the denominator of the sensitivity for each $p \in P$
from below by $\left( \sum_{p \in P} |p^T x| \right)^z$. This allows us to use the f-SVD with respect to the $\ell_1$-regression problem as done.

**Q**: Corrections are needed for related work. **A**: All suggested corrections were accepted and added to the related work,
we thank the reviewer for pointing this out.

**Reviewer 2:** We thank the reviewer for the constructive suggestions. We incorporate them in the new version as
explained below, and hope to raise the final scoring.

**Q**: Do we assume that g and h are given? **A**: As for the applications specified in our framework, $g$ and $h$ were established
theoretically, and we assume they are given. As part of our future work, we aim to generalize this framework as well as
having a polynomial algorithm for finding the proper $g, h$ for each "near-convex" function (see Definition 1).

**Q**: Moreover, some more explanation on finding matrices $D$ and $V$ would be helpful. **A**: We have added intuition and
explanation regarding this. Basically, we explain (in a more detailed fashion) that $D$ is a diagonal matrix where its
diagonal entries are proportional to the Löwner ellipsoid's axis lengths, while $V$ represent the basis of the ellipsoid.

**Q**: It might be helpful to discuss an example after Definition 1. **A**: Added for the logistic regression problem.

**Reviewer 3:** We thank the reviewer for his supportive comments and very helpful suggestions.

**Q**: Figure 1 is nice. It would be good if they bring it to the beginning of the paper and explain the intuition better. **A**:
We thank the reviewer for this very nice suggestion. We changed its place and explained the intuition better.

**Q**: They can sell better this nice results. **A**: We are not so good at selling, but hope that the strong result will help us
here. We thank the reviewer also for the selling suggestions.

**Reviewer 4:** We thank the reviewer for the supportive scoring and for the detailed review.

**Q**: The Figure captions are in a font that is too small!!!. **A**: We thank the reviewer for pointing this out. We changed the
fonts as requested.

[Meta-Review · NeurIPS 2020]

The paper cleans up several subtle loose ends in the analysis of coresets problem for Near-Convex Functions, in a nice unified way. This is mainly a theory paper, but some concerns on the experiments were raised that must be addressed by the author for the camera ready version. Please, make sure to incorporate the promised edits on the related work and improvements on the experimental part into the final version.